# Systematic assessment of the achieved emission reductions of carbon crediting projects

Benedict S. Probst [1,2,3] ✉, Malte Toetzke [1,4], Andreas Kontoleon [3], Laura Díaz Anadón [3,5], Jan C. Minx [6,7], Barbara K. Haya [8], Lambert Schneider [9], Philipp A. Trotter [10,11], Thales A. P. West [3,12], Annelise Gill-Wiehl [13] & Volker H. Hoffmann [2]

Carbon markets play an important role in firms' and governments' climate strategies. Carbon crediting mechanisms allow project developers to earn carbon credits through mitigation projects. Several studies have raised concerns about environmental integrity, though a systematic evaluation is missing. We synthesized studies relying on experimental or rigorous observational methods, covering 14 studies on 2346 carbon mitigation projects and 51 studies investigating similar field interventions implemented without issuing carbon credits. The analysis covers one-fifth of the credit volume issued to date, almost 1 billion tons of $CO_2$e. We estimate that less than 16% of the carbon credits issued to the investigated projects constitute real emission reductions, with 11% for cookstoves, 16% for $SF_6$ destruction, 25% for avoided deforestation, 68% for HFC-23 abatement, and no statistically significant emission reductions from wind power and improved forest management projects. Carbon crediting mechanisms need to be reformed fundamentally to meaningfully contribute to climate change mitigation.

Carbon pricing has become a central approach to mitigating climate change, though the operationalisation and geographic scope vary considerably[1]. Carbon pricing has taken three approaches: emissions trading schemes (ETS), carbon taxes and carbon crediting mechanisms. Carbon crediting mechanisms—the focus of this study—allow project developers to earn carbon credits through voluntary mitigation projects such as forest protection or renewable energy projects. These carbon crediting mechanisms are established and operated by international organisations, such as the Clean Development Mechanism

(CDM) and Joint Implementation (JI) established under the Kyoto Protocol[2,3], national or sub-national governments, such as California's Compliance Offset Program[4,5], and non-governmental entities, such as Verra and the Gold Standard Foundation[1,6–9]. Carbon credits are used in different ways: in compliance markets[4], countries and firms buy credits to meet targets under the Kyoto Protocol and Paris Agreement or to meet obligations under ETSs, carbon taxes, or the Carbon Offsetting and Reduction Scheme for International Aviation (CORSIA). In voluntary markets, governments, firms, non-governmental organisations or

[1]Net Zero Lab, Max Planck Institute for Innovation and Competition, Munich, Germany. [2]Group for Sustainability and Technology, ETH Zurich, Zurich, Switzerland. [3]Department of Land Economy, Centre for Environment, Energy, and Natural Resource Governance, University of Cambridge, Cambridge, UK. [4]Public Policy for the Green Transition, Technical University of Munich, Munich, Germany. [5]Harvard Kennedy School, Harvard University, Boston, MA, USA. [6]Mercator Research Institute on Global Commons and Climate Change, Berlin, Germany. [7]Priestley International Centre for Climate, School of Earth and Environment, Leeds, UK. [8]Goldman School of Public Policy, University of California, Berkeley, CA, USA. [9]Öko-Institut, Berlin, Germany. [10]Schumpeter School of Business and Economics, University of Wuppertal, Wuppertal, Germany. [11]Smith School of Enterprise and the Environment, University of Oxford, Oxford, UK. [12]Institute for Environmental Studies (IVM), Vrije Universiteit Amsterdam, Amsterdam, The Netherlands. [13]Energy & Resources Group, University of California, Berkeley, CA, USA. ✉e-mail: benedict.probst@ip.mpg.de

individuals buy carbon credits to meet voluntary goals, such as off-setting residual emissions. Other forms of results-based finance also create demand for carbon credits, in which governments and international organisations purchase carbon credits from mitigation projects that countries implement to achieve their goals under the Paris Agreement[1].

To assess the climate benefits of carbon mitigation projects, it must be verified whether projects are additional and whether emission reductions or removals have been conservatively quantified, permanent, and not double counted. Additionality refers to the principle that a mitigation activity would not have occurred without the revenue from the sale of carbon credits[7,10–12]. Conservative quantification refers to approaches that reasonably ensure that emission reductions or removals are not overestimated[10,11]. Non-permanence refers to the risk that the emission reductions or removals be reversed later on, for example through wildfires in forestry projects[10,11,13]. Lastly, double counting means an emission reduction or removal should be used only once to achieve a mitigation goal or target[10,11,14]. Next to these basic principles, several other aspects are commonly considered important for quality. This includes avoiding negative environmental and social impacts, such as impacts on biodiversity and local communities; appropriate distribution of mitigation benefits and carbon credit revenues; ensuring that carbon mitigation projects effectively contribute to achieving net zero emissions by mid-century and avoiding locking-in carbon-intensive technologies or practices; and adequate governance structures of carbon crediting programmes, including concerning transparency and third-party auditing[3,6,9–11,15].

Yet, carbon credits have come under considerable criticism due to growing evidence suggesting that many projects may significantly overestimate their emissions benefits or might not lead to actual emission reductions at all[2–9,15–22] and that some projects lead to environmental or social harm[9]. Carbon credits are issued based on standards developed by carbon crediting mechanisms. The quality of carbon credits hinges on the robustness of these standards and choices made by project developers. Potential issues compromising additionality and quantification include flexibility for project developers to pick favourable data or make unrealistic assumptions[2,3,6,8,9,23], adverse selection[4,13], and use of outdated data or inappropriate methodological approaches in the standards[2,3,6,9,11,15,16,24,25]. There is also considerable debate on the appropriateness of claims made in association with carbon credits and whether the use of carbon credits hinders or accelerates mitigation efforts.

In this paper, we focus our analysis on the two most basic principles of carbon credit quality: additionality and conservative quantification. More than a dozen studies have assessed the quantity of emission reductions that carbon mitigation projects likely achieved relative to the carbon credits issued, which we denote as the 'offset achievement ratio' (OAR, see Methods). While not all carbon credits are used for offsetting emissions, the private sector has become the largest source of demand for carbon credits[1], principally for offsetting. We complement studies that directly evaluated carbon crediting projects with studies that evaluated similar interventions without issuing carbon credits (which we call 'field interventions'). To search and synthesise the extant literature we rely on a conventional systematic review methodology[26], specifically the Population-Intervention-Comparator-Outcome (PICO) framework. We only include academic studies in our assessment that quantify the likely achieved emission reductions relying on experimental or rigorous observational studies (see Supplementary Table 1 for detailed inclusion and exclusion criteria).

We proceed with the following steps: First, we defined keywords to identify potentially relevant scientific studies across all project types listed in Table 1. Second, we used the artificial intelligence-supported systematic review tool Active Learning for Systematic Reviews (AS Review)[27] to filter for relevant studies among 64,993 studies identified in the first step (Supplementary Fig. 1 and Supplementary Tables 2 and 3 for search terms). Third, we filled in studies that artificial intelligence missed from our own previous work and additional article searches. Fourth, we downloaded the full text of the studies and then manually checked their relevance based on our inclusion and exclusion criteria (see Supplementary Tables 4 and 5 for included studies). Fifth, two researchers independently extracted quantitative estimates of the achieved emission reductions from the studies and other relevant aspects of the study detailed in our Codebook (which is part of the supplementary data). Lastly, we computed the OAR by either directly using the estimates from the article (if the study reported the likely achieved emission reductions and the volume of issued credits) or by deriving the likely achieved emission reductions from the underlying data from collected data on issued credit volumes for the analysed projects, and/or by combining estimates from multiple studies (see Methods). Overall, our sample contains 14 studies evaluating 2346 carbon mitigation projects and 51 studies investigating similar field interventions implemented without issuing carbon credits. These 51 studies contain evaluations of projects that did not issue carbon credits but are similar to carbon credit projects (e.g. evaluation of cookstove projects). We discuss these studies qualitatively to complement our discussion in the section 'reasons behind low OARs across project types.'

We synthesise the existing literature relying on experimental or rigorous observational methods, covering 14 studies on 2346 carbon mitigation projects and 51 studies investigating similar field interventions implemented without issuing carbon credits. Our analysis covers about one-fifth of the credit volume issued to date, almost 1 billion tons. We estimate that less than 16% of the carbon credits issued to the investigated projects constitute real emission reductions, though the OAR varies considerably across projects. Our assessment, therefore, documents substantial and systemic quality problems across project types. These quality problems stem from adverse selection, the ability of project developers to make unrealistic assumptions or pick favourable data and inappropriate methodological approaches. Carbon crediting mechanisms need to be reformed fundamentally to meaningfully contribute to climate change mitigation.

## Results and discussion
### Carbon mitigation projects and field interventions
To calculate OARs, we prioritised our search terms to cover the largest project types issuing credits through independent mechanisms and the Kyoto Protocol's CDM and JI (Table 1 and Supplementary Tables 2 and 3).

Credit issuance is concentrated among the Kyoto Protocol's crediting mechanisms as well as several governmental, private, and non-governmental mechanisms (which we collectively refer to as 'independent mechanisms') (Fig. 1a) and a few sectors (Fig. 1b). The CDM and the JI Mechanism have jointly issued 63% of credits (3.3 gigatons)[3,28], whereas independent mechanisms are responsible for 37% of issued credits (1.9 gigatons)[29]. Credits from chemical processes as well as industrial manufacturing projects were mainly issued under the Kyoto mechanisms, whereas credits from forestry and land use, as well as household and community projects, were mainly issued under independent mechanisms. Projects from renewable energy constitute 29% of the issued credits across these crediting mechanisms. Industrial manufacturing and chemical processes and account for 24% and 22%, respectively. Forestry and land use account for 15%, whereas waste management and household and community account for 5% and 3%, respectively. Domestic crediting mechanisms are excluded from this overview as these only constitute a minor fraction of issued credits[1]. Figure 1 displays historical averages and, therefore, current issuance volumes might differ.

Drawing upon the typologies of the Berkeley Carbon Trading Project[30], the CDM[28], UNEP DTU[31] and the Carbon Credit Quality Initiative[11], we classify each of the 65 studies in our assessment into

**Table 1 | Main sectors and project types of carbon crediting projects covered by our search terms**

| Forestry and land use | |
|---|---|
| Avoided deforestation | Activities designed to reduce deforestation. They are often based on a range of strategies such as improved monitoring, law enforcement and promotion of sustainable land-use practices |
| Improved forest management (IFM) | Applying practices which increase above and below-ground carbon stocks relative to the baseline, including by reducing timber harvest levels, extending timber harvest rotations, designating reserves, reduced impact logging, enrichment planting and stand irrigation or fertilisation |
| Afforestation and reforestation | Planting trees or reducing barriers to natural regeneration |
| **Renewable energy** | |
| Wind | Installing grid-connected wind power plants, replacing fossil-fuel-based electricity generation |
| Hydropower | Installing grid-connected hydroelectric power plants, replacing fossil-fuel-based electricity generation |
| Solar | Installing grid-connected solar power plants, replacing fossil-fuel-based electricity generation |
| Biomass | Installing biomass-fired power plants, including cogeneration plants, replacing fossil-fuel-based electricity generation |
| **Waste management** | |
| Landfill methane | Combustion of gas collected from solid waste disposal sites |
| Wastewater | Installation of less greenhouse-intensive wastewater treatment methods |
| **Chemical processes** | |
| $N_2O$ destruction in nitric acid production | Installing abatement measures to reduce $N_2O$ emissions from nitric acid plants |
| $N_2O$ destruction in adipic acid production[a] | Installing abatement measures to reduce $N_2O$ emissions from adipic acid plants |
| Hydrofluorocarbon (HFC)–23 destruction | Capturing and destroying HFC-23 produced as a waste gas from HCFC-22 production |
| Sulphur hexafluoride ($SF_6$) replacement and other | Avoiding $SF_6$ emissions by partial or full replacement of $SF_6$ cover gas with alternate cover gases, gas recycling or leak reduction |
| SF6 waste gas destruction | Capturing and destroying $SF_6$ waste gas streams in $SF_6$ production |
| **Household and community** | |
| Cookstoves | Distributing efficient cookstoves to households or institutions, reducing greenhouse gas (GHG) emissions by using less fuel, burning fuel more completely and/or switching to a less GHG-intensive fuel |
| **Industrial manufacturing** | |
| Mine methane capture | Flaring or combustion of gas captured from active and abandoned coal and other mines |
| Natural gas electricity generation | Installing new natural gas-fired grid-connected electricity generation plants, replacing fossil fuel-based electricity generation |
| Associate gas recovery[a] | Avoid flaring of associated gas in oil and gas production |
| Energy efficiency[a] | Improvement of energy efficiency in industry such as recovery and utilisation of waste heat |
| Avoiding uncontrolled fires from coal waste piles[a] | Avoiding GHG emissions from uncontrolled fires from coal waste piles, e.g. by extracting coal from the piles, leaving bare rock which does not ignite, or extinguishing the fires |
| **Carbon capture and storage** | |
| Carbon capture and enhanced oil recovery | Capturing carbon dioxide from industrial processes followed by compression, transport and injection for permanent storage underground while also enhancing oil recovery |

Based on the classification from the Berkeley Carbon Trading Project[30], the Clean Development Mechanism (CDM)[28], the Danish Technical University and United Nations Environment Programme (UNEP DTU)[31], and the Carbon Credit Quality Initiative (CCQI)[11].
[a]Please note that these project categories did not have specific search terms but were covered by the generic search terms (see Supplementary Tables 2 and 3, under 'generic').

one of seven sectors and one of the 21 project types listed in Table 1. We differentiate between studies investigating carbon crediting projects and studies investigating similar field interventions that were implemented without issuing carbon credits (which we refer to as field interventions). We found 14 studies investigating 2346 carbon crediting projects across six project types (Fig. 2a; please note that the sector forestry and chemical processes contain two project types, respectively) and 51 studies investigating field interventions without issuing credits with a total of 1.2 million observations (Fig. 2b). For the other three sectors (waste management, industrial manufacturing and carbon capture and storage), we could not find any studies investigating field interventions and carbon crediting projects that matched our inclusion criteria (see Supplementary Table 1). Overall, we find the strongest concentration of carbon project evaluations in the forestry sector, with equal distribution across the other sectors (Fig. 2a).

Studies on carbon credit projects are generally split between different geographies (Fig. 2c); Africa is an exception, with no studies focused solely on the continent but covered in three studies that evaluate multiple geographies. Similarly, most field interventions

focus on forestry mainly in Latin America, as most forestry projects have been implemented in the Amazon region (Fig. 2d). Overall, studies of both carbon-crediting projects and field interventions available in the literature mainly rely on rigorous observational studies (Fig. 2e, f). In contrast to randomised controlled trials (RCTs) (in which the experimenter assigns treatment)[17], rigorous observational studies build a plausible control group to estimate project impacts[8]. Only 9 of 65 studies were based on RCTs (mainly evaluating the impact of fuel-efficient cookstoves, with one study in forestry[32]).

## The offset achievement ratio across project types
Carbon project developers quantify emission reductions in line with standards and methodologies developed by carbon crediting mechanisms such as the Verified Carbon Standard by Verra. Following an audit by an accepted third party, carbon credits are issued into a registry[8]. Yet, these standards and methodologies vary in their robustness and often allow for activities to be credited that would have happened regardless of the offset programme[2,23], and provide flexibility to project developers to select methodological approaches and data that maximise credit issuance[6,9]. It is, therefore, critical to contrast

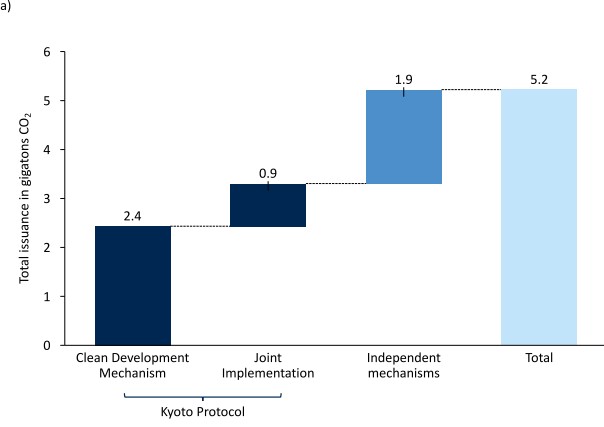

a)

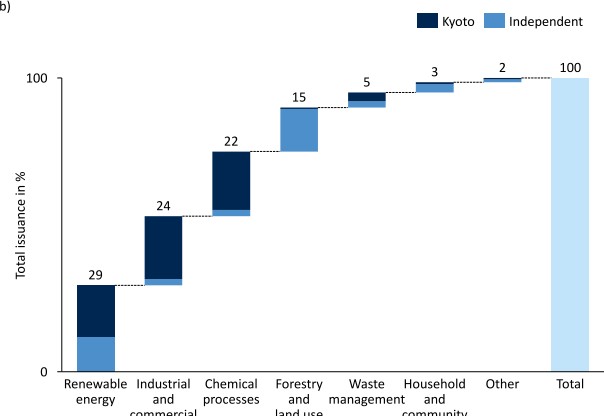

b)

**Fig. 1 | Overview of total issuance and relative share of sectors across all major crediting mechanisms. a** Total issuance in gigatons $CO_2$ across the Kyoto Protocol's two project-based mechanisms (Clean Development Mechanism and Joint Implementation) and the four major independent mechanisms covered by the Berkeley Carbon Trading Project Database[29] (American Carbon Registry (ACR), Climate Action Reserve (CAR), Gold Standard (GS) and Verified Carbon Standard (VCS)). **b** Total issuance in % across different sectors under Kyoto and independent mechanisms. Data is based on Clean Development Mechanism's (CDM) database for PAs and PoAs[28], the United Nations Environment Programme (UNEP) RISOE database[3] for JI and the Berkeley Carbon Trading Project Database v9[29]. Other crediting mechanisms are excluded as they only constitute a minor share of issued credits[1]. The sector 'other' contains carbon capture and storage, agriculture and transportation.

the emission reduction estimates used to determine credit issuance to those achieved based on rigorous academic assessments.

We introduce the term 'offset achievement ratio', which compares studies' quantitative estimates of carbon crediting projects' emission reductions with those made by project developers to generate carbon credits. An OAR of 50% indicates that the academic literature estimates that only half of the emission reductions claimed by project developers—and issued as carbon credits—were likely achieved. We complement these quantitative estimates with qualitative discussion of other studies including other qualitative and quantitative studies of the quality of offset methodologies and studies that assess field interventions that did not issue carbon credits but may still hold important insights on additionality, conservative quantification, or other relevant factors.

To quantify the OAR, we rely on academic studies that evaluate voluntary, project-based activities that seek to reduce emissions or enhance removals (see Supplementary Table 1 for inclusion and exclusion criteria). We excluded studies that evaluate non-voluntary activities such as mandatory regulations or non-project-based

activities (e.g. other forms of carbon pricing such as carbon taxes). We focus on studies that evaluate project impact against a credible comparator. This comparator can include projects, land, or households that were not part of the carbon crediting projects[4,5,7,8,17,21]; this can include historical data of the same project before it became a carbon crediting project[15,16]. The comparator can also be values from the scientific literature[6]. For example, some studies compare individual factors used by carbon crediting projects, such as the share of users that adopt a fuel-efficient cookstove, against the body of knowledge in the published literature[6]. Studies must also include a quantitative assessment of greenhouse gas emission changes or a comparable environmental metric, such as deforestation rates[7,8]. Lastly, we only include studies that use RCTs or rigorous observational data (which construct a plausible control group[8] or science-based comparator[6] to estimate project impacts). The included studies fall into several categories: peer-reviewed articles[17], papers aimed at peer-reviewed journals (e.g. working papers)[18] and chapters in PhD theses[19], which also undergo an academic examination process. We exclude qualitative studies from our quantitative assessment.

Our assessment considers additionality and conservative quantification in determining the OAR. The latter encompasses project, baseline and leakage emissions. Figure 3 illustrates which of these issues have been addressed by the 14 studies on carbon crediting projects that were considered in determining the OAR. Not all studies address all factors that affect a particular source of over-crediting. For instance, Aung et al.[17] studied the impact of fuel-efficient cookstoves on firewood usage in households that received the stove and those that did not (i.e. project and baseline emissions). However, the authors do not address other over-crediting factors related to the project emissions and baseline, such as the fraction of non-renewable biomass used to compute credit issuance. In contrast, Gill-Wiehl et al.[6] cover all relevant factors relating to over-crediting from baseline and project emissions.

## Offset achievement ratio across project types
Overall, we find that carbon-crediting projects achieved considerably lower emission reductions than the number of credits issued to the projects (Fig. 4). We find the lowest OARs in wind power in China and improved forest management (IFM) in the United States, for which no statistically significant emission reductions were documented in the studies (we, therefore, assume an OAR-value of 0% for these projects; see Eq. (1) in the Methods section, as well as Supplementary Table 6 detailing the exact numbers used to calculate the OAR across and within project types). These project types are followed by cookstoves (10.8%), $SF_6$ destruction (16.4%), avoided deforestation (24.7%), and HFC-23 destruction (68.3%) (Fig. 4a). Project-level results (Fig. 4b) show that individual projects may (over-) deliver relative to the issued credits, but the vast majority underachieves relative to the volume of issued credits. For our estimates in Fig. 4a, b, we use the central estimates from the studies. The source data are provided in this paper.

## The offset achievement gap
The studies in our assessment cover projects that are responsible for 19% of carbon credits issued across the main international and independent carbon crediting mechanisms (Fig. 5a). Using the OAR estimates, we find that of the 972 million credits issued across the covered project types, 812 million likely do not constitute real emission reductions (Fig. 5b). This offset achievement gap is larger than Germany's annual emissions. The largest source of non-achieved credits stems from avoided deforestation, wind power and IFM (Fig. 5c). Note that we only include credits from methodologies and projects that are covered by the underlying studies. For instance, we only include credits from Chinese wind power plants under the CDM[19] or IFM projects that use California's Air Resources Board protocol[4,5]. For avoided deforestation[7,8,21] and cookstoves[6,17], we apply the OAR to all credits

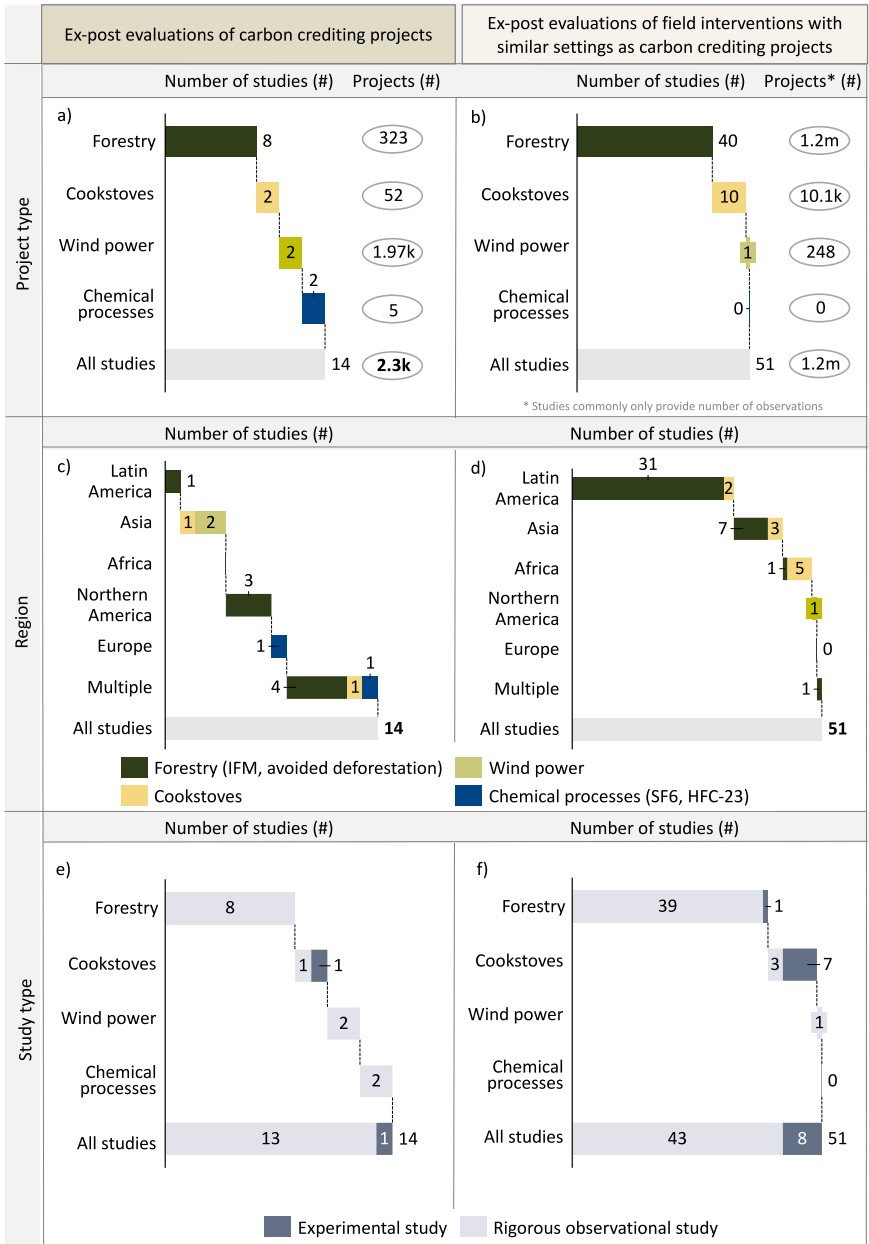

**Fig. 2 | Overview of studies in the systematic assessment. a/b** Distribution of studies across project types, (**c/d**) across regions and (**e/f**) methodology types. Note: k refers to the number in thousands, and m refers to the number in millions. See Supplementary Tables 4 and 5 for a full list of studies. Note: IFM refers to improved forest management. If the same project was evaluated by multiple studies, we count each as a separate project evaluation as the time frame, methodology and other relevant factors might differ. The total project number includes studies that could not be integrated into our quantitative framework (e.g. ref. 18) but are still discussed in the relevant sections.

issued to the project types using the studied methodologies, as the underlying studies cover a representative sample of projects.

**Reasons behind low offset achievement ratios across project types**

Overall, our assessment indicates that the total achieved emission reductions of the carbon crediting projects for which evidence is available are substantially lower than claimed. We discuss potential sources behind the offset achievement gap across the analysed project types.

**Improved forest management (IFM)**

Two studies[4,5] investigating 106 IFM projects did not find statistically significant reductions in carbon emissions and removals from IFM

activities under the ARB protocol. These studies focus on project emissions and baselines. IFM projects involve forest management practices that increase carbon in forests and/or reduce carbon loss in forests. While IFM activities can include extending harvest rotations, reduced impact logging, liberation thinning and converting logged forests into conservation forests, most IFM projects mainly generate credits from avoiding forest degradation. Globally, over three-quarters of carbon credits from IFM projects were issued under the California Air Resources Board's US Forest Projects Protocol[29], and the protocol has been the focus of the two studies of the quality of IFM carbon credits included in our assessment.

Stapp et al.[4] analysed 90 IFM projects and overall found no statistically significant evidence of additionality across the United States over the first 5 years of the projects when compared with control lands.

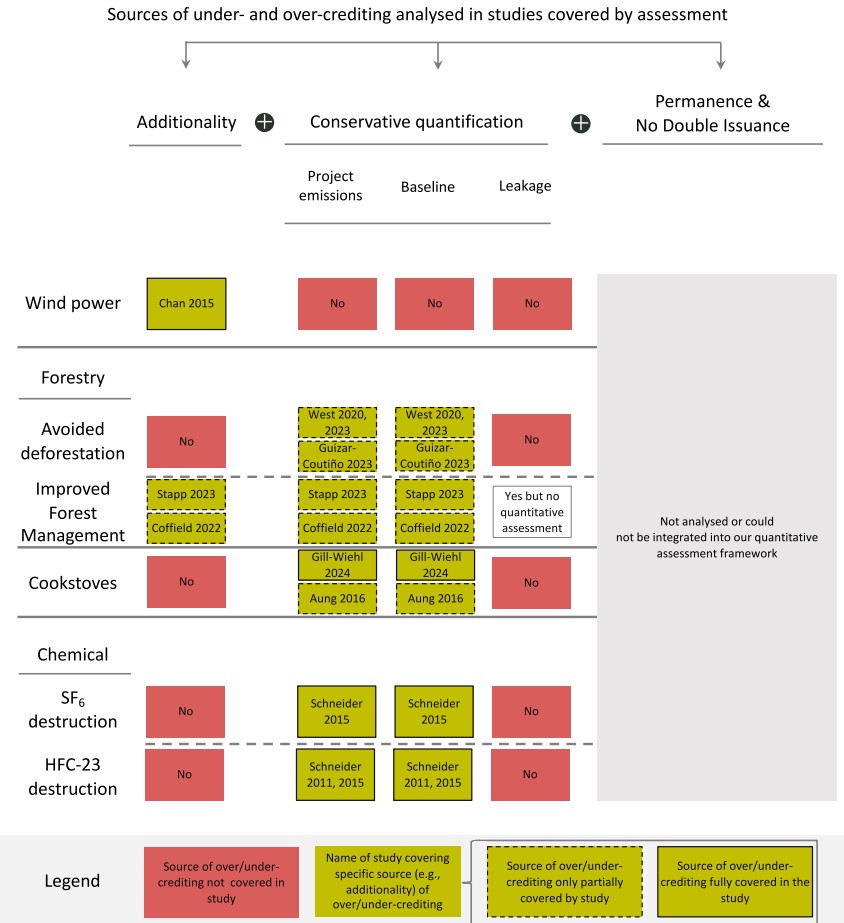

**Fig. 3 | Sources of under/over-crediting analysed by carbon crediting studies covered in our assessment.** The name of the authors shows the study, which analysed the specific source of over/under-crediting, otherwise, the box indicates 'No'. The figure excludes several studies that analyse offset quality, namely Calel et al.[18] Badgley et al.[33] Holm et al.[20] and Bomfim et al.[45], because they could not be integrated into our quantitative assessment framework, but the findings are reviewed in the discussion section. Reasons for exclusion for each of these studies can be found in Supplementary Table 7. Only the first author's name is shown due to space constraints. Note that the figure does not cover the field interventions as these did not issue carbon credits and, therefore, could not be integrated into our quantitative framework. IFM refers to Improved Forest Management.

The authors document heterogeneous impacts across sub-groups. They observe reduced harvests for land owned by timberland investment management organisations and real estate investment trusts but increased harvesting from other groups. Overall, the positive and negative effects on harvesting balance out across the study sample. The study explains this lack of impact as adverse selection. The baseline is often set as the average carbon per hectare for the forest type in the region of the project. The study finds that lands enroled into carbon crediting projects already had lower rates of harvest over decades before the start of the carbon crediting project compared to the average lands used to set the baseline. Hence, these projects were able to accumulate carbon compared to the baseline before the project started and then generate credits against an average baseline without needing to change how the forests were being managed.

Using a comparable approach, Coffield et al.[5] also find no evidence of additionality from 16 ARB IFM projects in California. The study found no statistically significant evidence of increased carbon accumulation after project initiation compared to similar control areas. Similarly, it found no evidence of reduced harvesting compared to past harvesting rates in the project areas and compared to harvesting rates of similar control areas. Lastly, while Badgley et al.[33] (72 projects analysed) could not be integrated into our quantitative assessment, the authors also document systematic over-crediting in California's carbon offset programme due to adverse selection.

Other studies of ARB IFM projects have found additional sources of over-crediting, suggesting that even if some projects changed their forest management practices, the emission reductions or removals would still likely be overestimated due to methods for assessing leakage[25] and for quantifying reversal risk and associated contribution of credits into the insurance buffer pool[13]. No studies to date have conducted quantitative assessments of the quality of credits under other IFM protocols. However, similar issues of lenient baselines, low leakage deductions and low deductions for reversal risk into the buffer pool have been documented for most protocols[24].

## Wind power

Two studies[18,19] investigated 1966 wind power projects registered under the CDM in India and China. These studies only investigate the additionality of these projects. Globally, around half of credits from wind power projects were issued under the CDM, 63% of which were generated in China. We use only the data by Chan and Huenteler[19] to estimate the OAR of wind power projects, because ref. 18 only identify the most obvious cases of non-additionality but provide no central additionality estimates for all projects.

Chan and Huenteler[19] investigated the additionality of 2051 wind projects, of which 1494 were financed in China under the CDM between 2007 and 2012. They found no statistically significant evidence that projects that received funding from the CDM were less

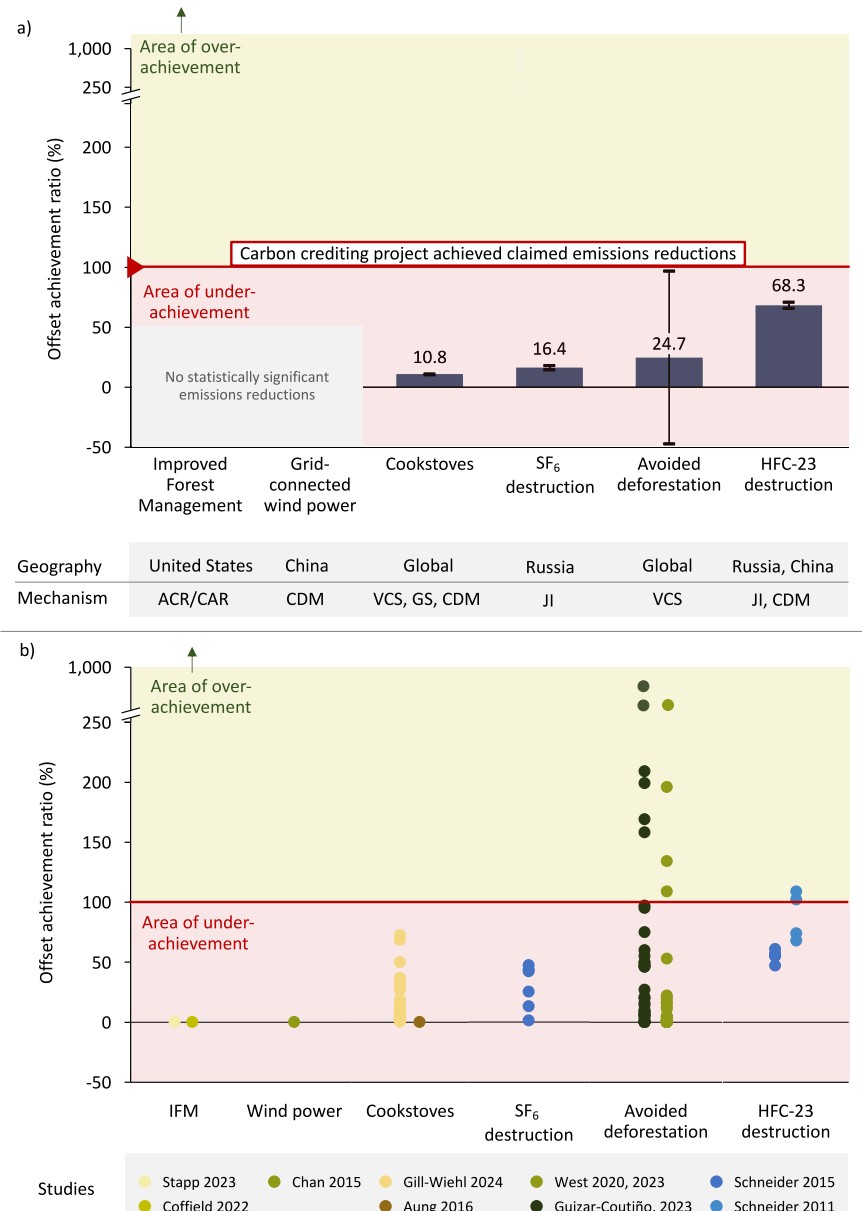

**Fig. 4 | Estimated offset achievement ratio of carbon crediting projects.**
**a** Estimated offset achievement ratio for project types for which we found relevant studies. **b** Project-level estimates extracted from relevant studies for individual projects. Only the first authors are mentioned in the study section for visualisation purposes. Two studies excluded that only showed upper-bound estimates refs. 18,35 and two studies that could not be integrated into the framework[20,45]; see Supplementary Table 7 for detailed exclusion reasons. Estimated average offset achievement ratios in (**a**) are the weighted average of projects' OAR based on the issued credits (i.e. projects that issued more credits are weighted more). Each dot in (**b**) represents one project-level OAR, with the colours corresponding to the underlying studies. Confidence intervals in (**a**) are the weighted variance (by credit issuance) based on the individual estimates in (**b**), whereas the centre of the error bar represents the weighted average for each project type. The exact issued credits and the OAR for each project can be found in the supplementary data. ACR refers to the American Carbon Registry, CAR to Climate Action Reserve, CDM to Clean Development Mechanism, VCS to Voluntary Carbon Standard, GS to Gold Standard, JI to Joint Implementation and IFM to Improved Forest Management.

financially viable than those constructed without support. However, they show that projects under the CDM used more foreign technologies and larger wind turbines, potentially increasing technology transfer. In addition, they document a small positive effect on CDM projects being sited in previously undeveloped areas. Yet, these positive effects can only be ascribed to CDM financing if projects were additional, which appears not to be the case.

Calel et al.[18] investigate the additionality of 1350 wind projects in India, of which 472 were financed under the CDM between 2000 and 2013. They developed a new conceptual framework called Blatantly Infra-marginal Projects, which identifies particularly obvious cases of non-additionality. The approach allows the authors to identify projects that were less financially attractive but were built even without selling carbon credits. For around half of these projects, they identified that these projects had lower capacity factors, were in less windy locations and were sited further away from electrical substations, and hence, overall, likely to be less financially attractive than the CDM projects.

The authors indicate that low additionality is likely due to the capital intensity of this project type. Utility-scale renewable energy projects require high up-front investments and a secure cash flow to secure funding from banks and investors[34]. As revenue streams from

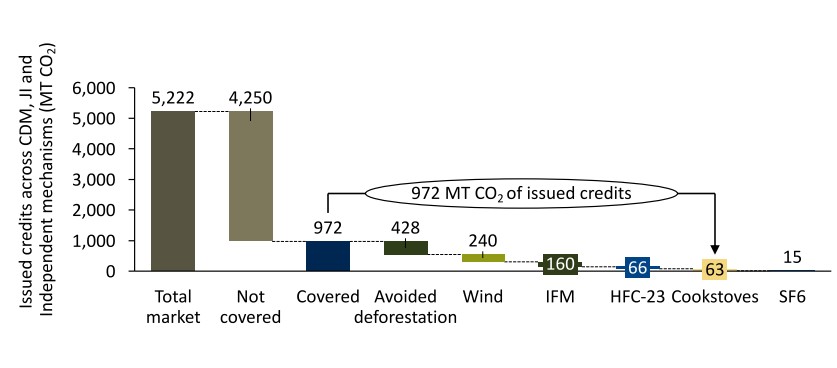

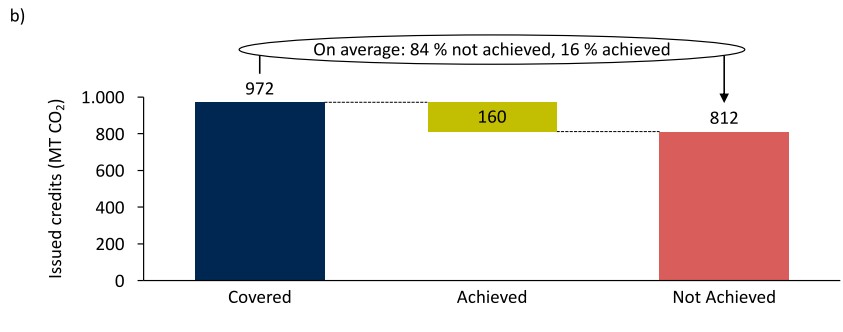

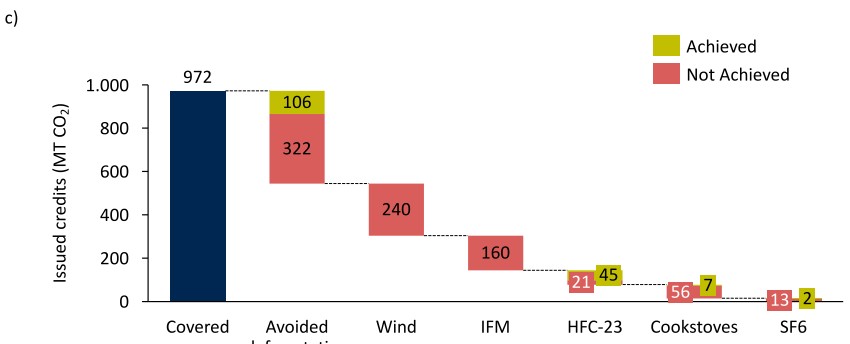

**Fig. 5 | Estimated offset achievement. a** Credits covered in our analysis relative to the total volume of issued credits based on sources from Fig. 1a. **b** Estimated achieved vs non-achieved emission reductions across covered projects based on percentage estimates from Fig. 4a. **c** Disaggregated shares of estimated achieved and non-achieved emission reductions across covered projects. IFM refers to improved forest management.

selling carbon credits are often low compared to revenues from electricity sales and carbon credit prices may fluctuate substantially, as in the CDM, revenues generated by carbon credits are unlikely to affect the financial viability of renewable energy projects substantially[19,23]. We do not assess small-scale projects, such as off-grid energy, due to a lack of studies.

### Cookstoves

Information from two studies[6,17] investigating 52 projects was used to estimate an average OAR of 10.8% (Supplementary Table 6 explains how we post-process and synthesise the results from these studies; this is the weighted average across projects covered by studies). Aung et al.[11] assess project and baseline emissions for one CDM project. Gill-Wiehl et al.[6] analysed 51 projects (40% of all issued credits across independent crediting mechanisms from five key methodologies) and assessed all relevant factors (apart from additionality and leakage) in the quantification of emission reductions, including fraction of non-renewable biomass, adoption/usage rates, and emission factors.

Distributing fuel-efficient cookstoves seeks to reduce greenhouse gas emissions by subsidising households in low- and middle-income countries to switch to a less GHG-intensive fuel or a more energy-efficient stove. Most cookstove projects are registered under the Gold Standard (GS), the VCS or the CDM and rely on GS and CDM[6] methodologies.

Aung et al.[17] ran an RCT to evaluate the climate impacts of one CDM-approved stove replacement project in India. The author team randomly assigned 187 households to either receive a fuel-efficient replacement (96 households) for their traditional stove or to serve as a control group. Overall, Aung et al. find no statistically significant impact on fuelwood usage between the intervention and control groups (hence, we assume an OAR of 0%). They document that 40% of households that received the fuel-efficient stove continued using the traditional stove. They hypothesise that the lack of reductions might also be due to households cooking larger meals with the improved stoves ('rebound effect'), thereby eliminating any efficiency-based reductions in fuelwood consumption.

While Aung et al. only analysed one project, Gill-Wiehl et al.[17] assessed the overall quality of a substantial portion of cookstove credits on the voluntary carbon market, covering 51 projects, five key cookstove methodologies and a comprehensive set of factors. The authors recalculate the likely emission reductions of these analysed cookstove projects by scrutinising key methodological assumptions made to issue credits. Overall, the authors find that the project sample likely only achieved 10.9% of the claimed emission reductions, though there is a large variation between methodologies (please note that the OAR of 10.8% calculated for the overall project type is the weighted average by issued credits from refs. [6,17]). For instance, Gold Standard's Metered methodology[35], which assesses fuel use directly, features the lowest over-crediting risks of all methodologies.

Hence, while efficient cookstoves have been found to offer considerable sustainable development benefits, the literature suggests that their low carbon credit quality is due to a lack of rigour and flexibility in how methodologies allow projects to (1) determine the fraction of non-renewable sources of fuelwood and other biomass (fNRB), (2) assess actual use of the new and old stoves and (3) translate these values into changes in fuel consumption. Only Gold Standard's Metered methodology accurately assesses stove use and fuel consumption by directly metering stove or fuel use. All other methodologies use methods with known biases or inaccuracies. To some extent, all use infrequent and simple surveys, which are vulnerable to bias when respondents give answers they believe the project developer wishes to hear[36]. Kitchen performance tests can have similar biases, when stove users change their behaviour when they are observed. Some methodologies also use stove efficiency ratings determined in laboratory settings that can be artificial and inapplicable to real-world conditions.

In addition, numerous other studies have evaluated one or a few factors in the emission reduction calculation and compared them to carbon crediting projects or methodologies' approaches, finding over-crediting from the choice of fNRB[37] and methods to track adoption/usage rates[38] and under-crediting from emission factors[39]. Rigorous evaluations of field interventions have found substantial variation in the achieved emission reductions[40–44], which are rarely on par with the levels claimed by carbon crediting projects[6]. Studies investigating the additionality and leakage of cookstove projects are still nascent in the literature but analysing these factors would be important to fully assess the achieved emission reductions[6].

## Avoided deforestation

Three studies[7,8,21] investigating 48 projects that seek to avoid deforestation were used to estimate an average OAR of 24.7% (see Supplementary Table 6 for description). For 26 projects, two independent estimates exist on their OAR (Fig. 6). Projects that seek to avoid deforestation employ various approaches, mostly to protect rainforests in the Global South, such as improved monitoring and control of deforestation in the areas and encouraging sustainable land uses[7]. All projects covered by our assessment that seek to avoid deforestation are registered under one of several VCS methodologies (e.g. VM0015, VM0007).

West et al.[7,8] investigated 36 projects (of which 32 projects contained sufficient data for analysis) across multiple jurisdictions and found an overall achievement ratio of 8.2%. The authors argue that a central reason for the low achievement ratio is the inherently flawed methodological frameworks used to calculate credit issuance. Specifically, project developers use deforestation baselines informed by historical trends in chosen reference areas defined at the outset of the project, which often result in unrealistic scenarios[7–9]. West et al.[7,8] recalculate the achieved emission reductions based on control areas not enroled in the project. Guizar-Coutiño et al.[21] investigated 40 projects (of which 35 contained sufficient data for analysis) and found

a higher average OAR (42%) for a partially overlapping set of analysed VCS projects as in West et al.

Yet, we found that studies diverge somewhat in their OAR assessments, even if the same offset project is analysed. For the 26 projects that were analysed by West et al.[7,8] and Guizar-Coutiño et al.[21] the weighted average OAR is 14.5% (with West estimating 10.5% and Guizar-Coutiño 18.5% for the fully overlapping set of VCS projects). The estimates are moderately correlated, with a correlation coefficient of $r = 0.4$ (Fig. 6). Several reasons could explain this divergence, such as differences in methodology, selection of control groups and pixel vs. area-based approach. The observed divergence underscores the challenge of estimating baselines and the OAR of avoided deforestation projects. Estimates are very sensitive to the creation of the control group, a non-trivial task due to the unobservable nature of these groups and the necessity of their construction via statistical methods. Overall, while the findings from West et al. and Guizar-Coutiño et al. diverge, they indicate that forest protection was much less effective than the volume of issued credits indicates.

Yet, West et al. and Guizar-Coutiño do not assess project developers' assumptions regarding the carbon contained in the forest areas, which can further lead to over-crediting (see Fig. 3). Bomfim et al.[45] assess project developers' estimates of the carbon per hectare in protected forests. If these estimates are overstated, then the issuance of credits will also be inflated. Based on a representative sample of 12 projects across four key VCS methodologies, the authors show that project developers have significant leeway in assessing carbon content in forests. They found that project estimates were 23%–30% higher than values drawn from scientific literature. We do not consider this potential additional source of overestimation in our OAR calculation, as more research would be needed to ascertain the carbon rates per hectare on a project level.

Further to the carbon crediting project evaluations, a large literature exists that assesses the effectiveness of interventions seeking to avoid deforestation or similar environmental degradation[46]. Studies have found a wide variance in the effectiveness of these interventions. For projects that have low performance, studies have documented various reasons, such as poor administrative targeting (i.e. the project does not protect the forest most at risk), adverse-self-selection (those without intention to deforest self-select into programmes) and non-compliance (many schemes do not have appropriate measures to sanction non-compliance)[46].

## Chemical processes

Based on two studies[15,16] evaluating HFC-23 and $SF_6$ projects in chemical processes, we derive OARs of 16.4% for $SF_6$ and 68.3% for HFC-23 destruction. These studies investigate project and baseline emissions but do not address leakage (see Fig. 3). The projects were registered under the CDM and JI.

Schneider[15] analysed assumptions about baselines made by 19 HFC-23 destruction projects under the CDM. For two projects (CDM 151, CDM 1105), the author observed monitoring periods in which the projects could not issue carbon credits due to methodological constraints. For these two projects, we leverage historical data and data observed in periods without carbon credit issuance to compute the OARs.

Schneider and Kollmuss[16] investigated four projects, three abating HFC-23 and/or SF6 under the JI mechanism in Russia and one trifluoroacetic acid (TFA) plant in France. We exclude the plant in France due to lacking historical data. To calculate the OAR for these plants, we follow a similar approach as in Schneider[15] (see Supplementary Table 6).

Generally, HFC-23 and $SF_6$ abatement projects have a high likelihood of additionality as there is commonly no business case for these interventions in the absence of financial or regulatory incentives. Yet,

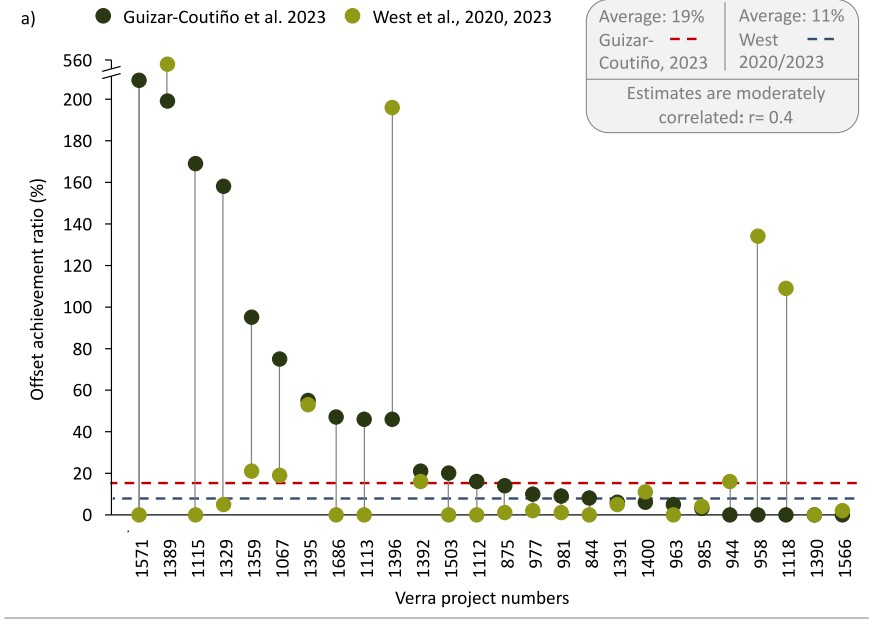

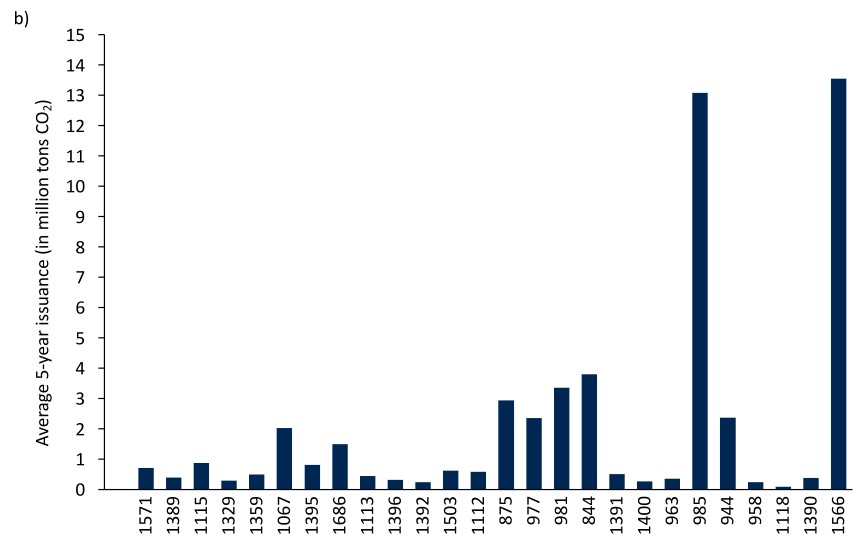

**Fig. 6 | Estimated offset achievement ratio of avoided deforestation projects analysed in two studies. a** Offset achievement ratio across studies. **b** Average 5-year issuance in million tons $CO_2$. Based on West et al.[7,8] and Guizar-Coutiño et al.[21]. Credit data is based on the VCS registry[50] and contains all credits that have been issued. Please note that for the main estimates in Fig. 4, we use the estimates from the full sample of projects, which is an OAR of 24.7% (compared to an OAR of 14.5% on average for this subset of 26 projects). Average 5-year issuance is based on the total issuance from project start year to 2024 (calculated by dividing total issuance by total years since inception, multiplied by five).

the high carbon credit revenues can lead to perverse incentives to increase waste gas generation beyond levels that would occur without carbon credits. The two CDM projects lowered their HFC-23 waste gas generation in periods when they could not claim carbon credits[15]. The CDM Executive Board revised the respective methodology to address this issue, but most plants never applied the new methodology as they stopped issuing credits due to a lack of demand. The HFC-23 and $SF_6$ projects under JI abruptly increased their waste gas generation at the point in time when plant operators could generate (more) credits by producing more waste gas[16]. For the HFC-23 projects, changes in waste gas generation were more moderate than for the two $SF_6$ projects for which waste gas generation also exceeded Intergovernmental Panel on Climate Change (IPCC) default values by up to 85 times, leading to lower OAR values for the $SF_6$ projects compared to the HFC-23 projects.

Next to these two studies that qualify for our analysis, several studies have assessed the quality of projects abating nitrous oxide ($N_2O$) from adipic acid and nitric acid production[2,3,22]. These studies indicate that carbon leakage may have led to some over-crediting from CDM projects abating $N_2O$ from adipic acid production. For $N_2O$ abatement from nitric acid production, older CDM methodologies (AM0028 and AM0034) involve considerable uncertainty regarding $N_2O$ generation in the baseline and pose some risk of over-crediting, whereas a new methodology version (ACM0019) is likely to lead credit fewer emission reductions than are actually occurring.

**Implications for carbon crediting mechanisms**
We synthesised the extant literature relying on experimental or rigorous observational methods, covering 14 studies on 2346 carbon mitigation projects and 51 studies investigating similar field interventions

implemented without issuing carbon credits. Our analysis covers about one-fifth of the credit volume issued to date, almost 1 billion tons. We estimate that less than 16% of the carbon credits issued to the investigated projects constitute real emission reductions, with 11% for cookstoves, 16% for $SF_6$ destruction, 25% for avoided deforestation, 68% for HFC-23 abatement and no statistically significant emission reductions from wind power projects in China and IFM projects in the United States.

Our assessment, therefore, documents substantial and systemic quality problems across all analysed project types, which further strengthens the evidence by previous cross-cutting analyses of the CDM and the JI[2,47]. Carbon credits are issued based on standards developed by carbon crediting mechanisms. The quality of carbon credits hinges on the robustness of these standards, the choices made by project developers in applying these standards and the thoroughness of the checks by third-party auditors and the carbon crediting mechanism. Our assessment highlights that many project developers pick favourable data or make unrealistic assumptions[6]. Some methodologies make use of outdated data or inappropriate methodological approaches[4], which can lead to adverse selection[35] or perverse incentives[12,15,16]. Our results also indicate that there is substantial heterogeneity across project types and methodologies.

The reviewed studies suggest that existing approaches to assess additionality have led to many non-additional projects being registered. To address this issue, carbon crediting programmes could limit eligibility to project types that have a high likelihood of additionality and of being effectively supported by revenues from carbon credits. For example, following criticism regarding additionality, Verra and the Gold Standard excluded wind power projects in most countries from eligibility. However, newer crediting mechanisms, such as the Global Carbon Council, include these projects in their scope. This change would result in a much narrower set of eligible project types.

Our findings also suggest that the standards and methodologies to quantify emission reductions need to be considerably improved. Such improvements should address a range of issues, in particular reducing project developers' flexibility in making favourable methodological assumptions to maximise credit generation[6,8,21]; using conservative assumptions and data based on the latest scientific evidence[6,18,19,45]; and addressing the risk of adverse selection[4,5] and perverse incentives[15,16]. Carbon crediting programmes may also exclude project types from eligibility where it is very difficult to ascertain whether calculated emission reductions result from the mitigation activities or exogenous factors that impact emissions, an issue that has also been referred to as 'signal-to-noise' issue.

Various other studies, not included in our analysis, suggest that quality issues also persist for many other project types not covered by our analysis[2,3,11,47,48]. Our estimate that 812 million carbon credits do not represent actual emission reductions should, therefore, be considered as a lower bound as many more credits currently traded may not constitute real emission reductions.

In addition, questions around additionality and leakage remain only partly addressed by the literature[24,25] and our analysis does not cover two other potential sources of over-crediting: permanence and double counting. For instance, Holm et al.[20] assess the non-permanence risk for 57 VCS forestry projects. Project developers need to make non-permanence risk assessments which inform the number of carbon credits set aside to insure against future reversals. Holm et al.[20] recalculate the assessments made by project developers based on the latest scientific literature and find that project developers were issued on average 26.5% more credits than an appropriate risk management would demand. Cookstoves projects also face a non-permanence risk as more fuel-efficient cookstoves lead to the preservation of carbon stocks in surrounding forests, but this risk is not accounted for by any of the carbon crediting mechanisms. Double issuance presents another risk as more than half of cookstove projects are co-located in areas where projects seek to avoid deforestation[49].

Hence, our estimates would likely be even lower if these factors were considered.

Our findings also suggest that more research is needed to better understand the quality of credits across different project types. For instance, for renewable energy, the extant literature providing quantitative assessments of achieved emission reductions focuses primarily on grid-connected wind power projects[18,19], though the literature on small-scale renewable energy is scant. More work is also needed to explore the full sources of over/under-crediting of projects with existing evaluations.

Demand for carbon credits is expected to grow significantly over the next decades, with increased demand from voluntary carbon market buyers, domestic compliance markets, CORSIA and countries using Article 6 of the Paris Agreement[1]. Yet, our results substantiate doubts about the environmental quality of carbon credits from the project types we study. These quality issues need to be addressed for carbon crediting mechanisms to meaningfully contribute to climate change mitigation.

## Methods
### Literature search
We follow ref. 26 in structuring and reporting the methods as well as their approach to searching the literature. To collect the relevant studies for our assessment, we proceeded as follows: (1) We perused existing reviews on credit quality and the studies cited in the reviews, which are all from the grey literature[2,3]. (2) We then searched the most prominent databases (Web of Science, SCOPUS) with keywords (Supplementary Tables 2 and 3). (3) Lastly, the author team also did individual searches on Google Scholar and perused the reference list of studies included in the review. All studies were downloaded on the 26th of August 2022. During the assessment process, we reran the search to see whether any relevant research had been published in the meantime and did manual searches to complement the existing set of studies.

To search existing bibliometric databases, we developed search strings. In line with a large body of systematic reviews, we employ the PICO framework to define keywords and search literature databases for our systematic assessment. The PICO framework includes the definition of a central research question, inclusion and exclusion criteria to select studies from the large pool of potentially relevant studies and a description of the final sample. The central question of this analysis is: 'What is known from the scientific literature about the differences between the emission reductions likely achieved by carbon crediting projects relative to the number of carbon credits issued?'

To operationalise the research question, we developed the search strings iteratively. We started with relevant studies known to the author team to define keywords. We searched for academic studies that evaluate voluntary, project-based activities that seek to reduce emissions or enhance removals. We excluded studies that evaluate non-voluntary activities such as mandatory regulations or non-project-based activities (e.g. other forms of carbon pricing such as carbon taxes). We focus on studies that evaluate project impact against a credible comparator. This comparator can include projects, land, or households that were not part of the carbon crediting projects; this can include historical data of the same project (e.g. a chemical plant producing waste gas before it became a carbon crediting project[9]).

The comparator can also be derived from the scientific literature. For example, some studies compare individual factors used by carbon crediting projects, such as methane oxidation rates at landfills or life-cycle emissions from charcoal production, against the body of knowledge in the published literature. Studies must also include a quantitative assessment of greenhouse gas emission reductions or a comparable environmental metric, such as reduced deforestation rates. The central feature of rigorous academic studies is that they

include a credible comparator and cross-check assumptions made by project developers with those based on the latest science.

The last criterion is that we only include studies that use RCTs or rigorous observational methods (which may include both modelling and empirical studies). RCTs or rigorous observational studies fall into several categories: peer-reviewed articles, working papers[13] aimed at peer-reviewed journals (i.e. pre-prints) and chapters in PhD theses[14], which also undergo an academic examination process. We exclude qualitative studies unless they are incorporated into a quantitative assessment.

We proceeded as follows. After having defined the keywords and inclusion and exclusion criteria, we ran the search on SCOPUS and Web of Science as well as manual searches on Google Scholar. As our keywords were inclusive and we did not impose restrictions based on the scientific discipline, publication date or study design, our search led to a large set of potentially relevant studies (64,993). After removing duplicates, our search returned 46,108 studies. We relied on the AI-supported systematic review tool AS Review[9] to order our complete study set in order of probable relevance. AS Review is a software tool that allows for more efficient screening of titles and abstracts. By labelling a set of potentially relevant articles AS Review prioritises articles to be investigated for relevance in the screening process. Screening articles without prioritisation is error-prone and inefficient as only a small fraction of articles is relevant.

A team of two researchers manually screened and labelled (relevant/not relevant) the title and abstract of the first 4,611 studies ordered by probable relevance based on AS Review (Version 0.18). While we cannot rule out that our prioritised screening approach omitted some relevant studies, our screening approach is in line with other AI-supported systematic reviews[26]. To ascertain that we did not miss critical studies, the author team also searched manually for relevant studies. Out of the assessed studies, 150 studies were flagged for full-text review. Of these 150 studies, 97 were excluded after critical appraisal due to non-relevance (31), absence of a credible comparator (56), no effect size (4), review (1), and percentage change reduction not reported (5). The final set included 65 studies, of which 12 studies were added based on a manual search. Then, two researchers independently extract the reported effect sizes from individual projects and other relevant aspects of the study detailed in our Codebook. For field interventions that did not involve the issuance of carbon credits, we discuss the findings qualitatively in the discussion section. In total, our final sample comprises more than 2000 carbon-crediting projects and 65 studies. The detailed ROSES flow diagram for systematic reviews can be found in Supplementary Fig. 1 and all included studies in Supplementary Tables 4 and 5.

## Calculating achieved emission reductions

The central goal of our assessment is to quantify the achieved emission reductions of carbon crediting projects relative to the issued credits, which we call the OAR. Not all carbon credits are used for offsetting (results-based climate finance and contribution claims made by companies are other uses), but since most credits are used for offsetting, we use the term OAR as Eq. (1):

$$OAR = \frac{A \times I}{C} \qquad (1)$$

where $C$ is the number of carbon credits issued to the project, A is the additionality factor with $0 \leq A \leq 1$ (0 indicating no additionality and 1 full additionality) and $I$ are the greenhouse gas emission reductions or removals expressed in tonnes of $CO_2$ equivalent achieved by the project.

$I$ can be further disaggregated in Eq. (2) into:

$$I = B - (P + L) \qquad (2)$$

where B is the baseline emissions, P is the project emissions and L is the leakage emissions in tonnes of $CO_2e$. Note that we omit other relevant factors for project impact here, such as permanence and double counting, which can further affect $I$. Also, if project emissions equal baseline emissions, the project has no impact (while holding leakage constant). The methodological approaches to estimate baseline, project and leakage emissions depend on the project type. For instance, for cookstove projects, key factors are the number of days a stove is in use, usage rates, the fraction of non-renewable biomass and the efficiency of the old and new stove.

To extract and standardise the estimates from individual studies, we first differentiate between carbon-crediting project studies and those that do not issue credits. For field interventions that did not issue carbon credits, we only qualitatively discuss the results in the discussion section. For carbon crediting projects, we extracted and commonly did further analysis to integrate the results into our quantitative OAR framework (see Supplementary Table 6).

For the carbon crediting projects, the metrics we are interested in are the additionality A and the achieved emission reductions $I$ that a project achieved relative to the issued credits. However, most studies do not report $I$ but a metric correlated with emission reductions, such as deforestation rates for avoided deforestation projects, harvesting and disturbance rates for IFM projects, and biomass use for cookstoves projects. In these cases, we transform the results into emission reductions (apart from studies for which no difference can be observed between carbon crediting projects and control groups, e.g. if a cookstoves project did not lead to reductions in biomass consumption, such as Aung et al.[17] the overall project likely had no impact on emissions). In our sample, the transformation only becomes relevant for projects seeking to avoid deforestation and chemicals (as the studies on other project types covered by our assessment either directly report achieved $CO_2e$-emission reductions or find no additionality or emission savings; see Supplementary Table 6 for a description).

For projects seeking to avoid deforestation, we compute how changes in deforestation rates between the project area and control areas translate to hectares of land prevented from deforestation and then multiply this quantity with the carbon stored per hectare (as reported directly by projects) in Eq. (3):

$$C_{total} = R_{deforest} \times C_{ha} \qquad (3)$$

where $C_{total}$ = Achieved credits by the project, $R_{deforest}$ = Total number of hectares prevented from deforestation based on academic evaluation, $C_{ha}$ = Carbon stored per hectare. To compute the OAR, we then divide the achieved credits by the number of issued credits, see Eq. (1).

Where separate studies on different quality elements were hard to combine, we focus our analysis on the most important factor as a lower bound of over-crediting and then describe the results of the other studies in the discussion section. For instance, as outlined in the discussion section, Bomfim et al.[45] evaluate project developers' estimates of the carbon per hectare in protected forests. The authors found that project estimates were 23–30% higher than values drawn from scientific literature. Yet, we do not consider this potential additional source of overestimation in our OAR calculation, as more research would be needed to ascertain the carbon rates per hectare on a project level.

For chemical processes, we use data on the historical waste gas generation to compute the OAR. We use data from periods prior to carbon crediting, periods in which the plants were not eligible for crediting or in which they could not claim more credits from increasing waste gas generation. In the case of $SF_6$ waste gas abatement, the study also compares the observed waste gas generation with default values from the IPCC. We use the average value from these scenarios as

the central estimate determined in Eq. (4):

$$OAR = \left( \frac{\text{Likely waste gas production without crediting}}{\text{Waste gas production during crediting}} \right) \quad (4)$$

We then synthesise the individual, project-level estimates into our central estimates presented in Fig. 4 via Eq. (5) for each project type (e.g. avoided deforestation, cookstoves):

$$OAR_{pt}, \text{weighted} = \frac{\sum_{i=1}^{n}(OAR_{it} \times C_{it})}{\sum_{i=1}^{n} C_{it}} \quad (5)$$

Where $OAR_{it}$ is the Offset Achievement Ratio for project $i$ over study period $t$ (if not otherwise specified), $C_{it}$ is the number of carbon credits issued to project $i$ over the study period $t$, $n$ is the total number of projects within the carbon project type $pt$.

### Reporting summary
Further information on research design is available in the Nature Portfolio Reporting Summary linked to this article.

## Data availability
The underlying studies and data are reported in Supplementary Tables 4, 5 and in our supplementary data. The underlying data of the study can be explored in our interactive online tool carboncredits.fyi, which we seek to regularly update with the latest study results. Source data are provided with this paper.

## Code availability
All relevant code can be found in the accompanying Excel (Version 16.88) supplementary data file.

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

## Acknowledgements

We thank Anna Neumann and Arber Sejdiji for research assistance, Niklas Stolz and Axel Michaelowa for helpful comments on earlier versions of this manuscript, and various experts who emailed us constructive feedback.

## Author contributions

B.S.P. designed and led the study's implementation. B.S.P. and M.T. developed the methodology with support from J.C.M., and B.S.P. and M.T. screened the studies. T.A.P.W. was responsible for the analysis of avoided deforestation, L.S. for chemical processes, B.K.H. for improved forest management, B.S.P. and P.A.T. for wind, and B.K.H. and A.G.W. for cookstoves. B.S.P., M.T., A.K., L.D.A, J.C.M., P.A.T., L.S., A.G.W., B.K.H. and V.H.H. edited the final manuscript.

## Funding

## Competing interests

The authors declare the following competing interests: L.S. is a member of the Executive Board of the Clean Development Mechanism. The other authors declare no competing interests.
