## [Peer Review File · Nature Communications]

REVIEWER COMMENTS

Reviewer #3 (Remarks to the Author):

Summary:

This article explores to what extent greenhouse gas offsetting projects are actually achieving emission reduction. To do so, the authors synthesize data from over 2,000 offsetting projects, categorize the projects into different offsetting associated sectors (e.g., forestry & land use), and estimate the average true volume in emission reduction per sector. Stunningly, the authors find that offset projects curtail much fewer emissions than the projects claim.

Overall, this is an ambitious research project that provides initial estimates of the effectiveness of carbon offset projects. Moreover, it documents differences between the offsetting sectors, and highlights aspects often neglected in offsetting projects, which is clearly of interest for the audience of Nature Communications. Personally, I think that the results could stimulate an important discussion about the credibility of carbon credits.

Despite this, I have several concerns regarding this article in its current form. I think most of the issues can be resolved.

Introduction:

- It is not clear what is meant with “field interventions” when the term is introduced in the introduction section (Line 87). I would provide the audience with a brief description of what is considered a field intervention (the description in the SI helped me a lot in that regard).
- What is the problem when the real offsetting reduction is dramatically overestimated? Please provide some implications for society so that the audience knows why this paper is relevant to the general public, which is currently missing in the introduction section.

Methods:

- By now, I do not sufficiently understand on how the true average carbon emissions reductions are estimated in Figure 3. I would add equations to the method section that sheds light on this.
- Next, I have a substantial concern regarding whether the estimates are unbiased (i.e., of Figure 3). In meta-analyses, one typically controls for differences in the studies. For instance, one could use a fixed effects or a random effects model. I think a random effects model would be more meaningful in this analysis, since it can substantially account for heterogeneity in the studies (e.g., sample number, subjects) that affect the calculated true offsetting reduction. This, in turn, is also helpful, since one could derive confidence intervals for the different sectors, acknowledging the uncertainty in heterogeneity (see, for an example, Möser & Bamberg, 2008 or Whitburn et al., 2019). Similarly, I think that confidence intervals—that one can extract from random effects models—are more meaningful than reporting the upper limits that the studies mention. Why? I would prefer an estimation of an upper limit average effect, considering the uncertainty in the studies. In doing so, you could also compare the point estimates of the different sectors by testing for statistical difference.
- To evaluate the true effect of field interventions, the authors pick other projects that are suitable for comparison to the respective field intervention. To do so, the authors randomly shuffle the list of

offsetting projects (Line 621). Later, when evaluating an intervention, suitable projects are taken from this list. My concern here is that it's not clear how the randomization has influenced the results and whether they are stable when shuffling the list again. I would expect some sensitivity analysis demonstrating that the results are consistent among different randomization runs.

- I absolutely share the view of the authors that it is necessary to control for “natural” changes in the absence of offsetting projects. So, changes associated with offsetting projects (e.g., higher efficiency due to new cook stoves) may happen independently of offsetting projects over time.

Still, I would be nice to have an estimate of the emission savings ignoring such natural changes (like most offsetting projects do). I suppose that estimate would make it really tangible to the audience why a “clean” calculation (considering ex-post counterfactual reduction), like you have presented, is needed. At the same time, this might also help to make the results more credible. For instance, by now, it is not clear to me why renewable energy projects are said to have 0% actual carbon reduction (Figure 3). To what extent is this influenced by taking the “ex-post counterfactual reduction” perspective that you introduce in this manuscript?

Minor:

- I do not understand “reduce deforestation from deforestation” in the introduction section (Line 100). Please revise.
- Line 542: To me, it is vague what “similar, real-world projects” could be here. Provide some specifications when projects are similar.

References:

- Whitburn, J., Linklater, W. and Abrahamse, W. (2020), Meta-analysis of human connection to nature and proenvironmental behavior. *Conservation Biology*, 34: 180-193. <https://doi.org/10.1111/cobi.13381>
- Möser, G., & Bamberg, S. (2008). The effectiveness of soft transport policy measures: A critical assessment and meta-analysis of empirical evidence. *Journal of Environmental Psychology*, 28:1, 10-26.

Reviewer #4 (Remarks to the Author):

This manuscript addresses an important issue in the field of carbon offset projects. In the recent past, particular attention has been paid to offset projects in the forestry sector. This manuscript makes a significant contribution by providing a scientifically sound analysis of the actual emission reductions of projects in different sectors. A quantitative approach is used to show that the emission reductions claimed by projects are not actually achieved.

The article complements existing studies with a novel scientific approach. The results show the need to fundamentally rethink carbon offset projects.

The scientific method used is sound and fully meets the expected requirements. The description of the scientific methods allows the traceability. In my opinion, there are no fundamental flaws.

In some places the manuscript should be revised.

Line 80: the review tool AS Review should be described briefly, possibly in the methods section.

Line 142 ff: obviously the reference to the figures is not correct. Here reference is made to figure 2 and not to figure 1.

line 170 ff. It should be described in more detail which biophysical and socio-economic characteristics are subject to the comparison. Forests differ in terms of tree species composition, site factors (e.g., climate, soil, topography), previous management activities, biomass volume, and other critical factors. The limitation to the shown aspects forest type, distance to roads and distance to forest edge) are not sufficient for a selection of reference areas.

line 312-347: a major problem in surveying carbon stocks in forests is the accuracy of the survey method. This should be addressed because the uncertainties that measurement, reporting and verification systems are subject to can have a decisive influence on the quantification of carbon stocks (see e.g. <https://doi.org/10.1016/j.ecolecon.2019.106513>)

Line 576: CBL instead of BL?

Line 586 ff: It is unclear whether the average values shown for the effectiveness index will be used for further study. If so, the choice of average values could mask extremes in individual projects.

Line 606-607: E is not defined. Should it be EI?

Line 624: It would be interesting to get an example from forestry projects, instead of the rather straightforward example of cooking stoves.

Reviewer #5 (Remarks to the Author):

In general, this is a well-written manuscript. The authors attempted to reveal the implementation efficiency of carbon offset projects across the main sectors that have been thoroughly discussed by previous studies. For a bibliometric study, it always has a cutting-edge perspective, together with methodological challenges. With no exception, I have following concerns towards this manuscript too:

First and most importantly, I understand it is very difficult to obtain actual, real data regarding ex-ante and ex-post carbon offset levels/volume. However, according to the authors' claim, "the offset achievement ratio' is the share of achieved emissions reductions based on credible academic studies relative to the claims made by project developers' ex-ante". I did not see many details here about how the authors defined "credible academic studies" and how these studies would provide plausible results that can be used to match and compare with projects. Therefore, such a ratio appears to be a subjective judgment based on a subjective material and some results are skeptical such as "0% for renewable energy" which is counterfactual. Unless the authors provide a strong defense, the true value of this manuscript is less likely to be explored.

Second, the discussion of leakage, durability, and co-benefits is important, but such a discussion appears to be inconsistent with the main focus of this study i.e. the carbon offset mismatch.

Third, I would not say the discussion is not logic or the evidence provided by the authors is not rich. However, this is again, a commonly seen issue for a review style study. The manuscript is entitled "systematic review", but in fact the authors attempt to further find out some reasons why there is a gap between proposals and reality. This is a pure economic logic (i.e., data, model and analysis) which is different to be presented in this type of studies thus it is not surprising that to some extent, the discussion is not quite comprehensive, even sometimes the authors say, "we could only find two

empirically rigorous studies...”.

Finally, the conclusion is too simple. The authors should not only state that they identified a gap, but they should also provide solid references and materials towards research directions in future. What should the academic world do for bridging such a gap?

Point-by-point response to referees

Dear referees,

Thank you for your constructive reviews on our submitted manuscript, which is now titled "*Systematic review of the achieved emission reductions of carbon-crediting projects*" due to the wider scope of our analysis and framing. We very much welcome the opportunity to submit a thoroughly revised version of our manuscript together with this point-by-point revision letter.

We highly appreciate the detailed and extremely helpful feedback from the reviewers. In the following, we would like to summarize the most important changes that we incorporated during the revision. We are confident that these changes address all major concerns. In addition, we strongly believe that they led to significant improvements.

1. **Revision of sub-sections by leading scholars:** We expanded the original author team so each subsection could be revised by a key scholar behind the relevant literature that we cover. Adding these authors is critical for vetting our previous analysis and to properly conduct post-processing of the original studies to make them comparable across project types and sectors. While our results have not fundamentally changed, we have added

substantial nuance to the findings, estimates, discussion, and conclusion as requested by all reviewers.

2. **Refinement of overall framing:** Based on comments by reviewer 3, we have reframed the introduction to detail the wider societal implications. The original paper focused narrowly on the voluntary carbon market, but most carbon credits were issued under the Kyoto Protocol (see Figure 1, main manuscript). We have, therefore, reframed the title, introduction, and general direction of the paper to include a broader set of crediting mechanisms (like the Joint Implementation Mechanism) and uses of carbon credits (offsetting, results-based climate finance, carbon taxes).
3. **Explicit conceptual framework:** We have developed a conceptual framework based on the extant literature that identifies all potential sources of under/over-crediting from carbon crediting projects. This framework elaborated in Figure 3, contains additionality, conservative quantification of emissions reductions/removals, permanence, and avoidance of double counting). This framework allows us to identify the exact sources of under/over-crediting that are analyzed in the studies, and which factors have been omitted in the extant literature.
4. **Reliability of data and estimation:** As suggested by Reviewer 3, we have now made explicit the underlying estimations used to calculate the offset achievement ratio (OAR) across sectors. For instance, Figure 4a now contains both the average OAR across project types and the individual project-level estimates (Figure 4b). The Figure also contains the exact geographic scope, crediting mechanism, and underlying studies (identified by colour codes). In addition, Supplementary Table 7 contains exact estimates and weight (based on analyzed credits) of the included studies.
5. **Discussion and findings.** As suggested by Reviewer 5, we have fully updated the discussion section, which is now firmly grounded in the findings from our study. We used the conceptual framework mentioned in 3) to structure the discussion. We show that existing studies focus on additionality and conservative quantification, but permanence and double counting remain mostly uncovered. As suggested by Reviewer 5, identifying these blind spots in the literature is critical for guiding future research.

We believe that our work on carbon credits is highly relevant to ensure effective Net Zero transitions. Given the tremendous growth of the use of carbon credits in corporate and national climate strategies, we believe that our critical assessment will help to better distinguish the quality of carbon credits and, thereby, help to improve the quality and credibility of these markets.

Based on the feedback of the reviewers, we have substantially improved the rigour, validity, and transparency of our method, expanded the scope and value of our analysis, as well as the contextualization and discussion of our results. We are confident that based on these changes the manuscript has improved considerably.

REVIEWER COMMENTS

Reviewer #3 (Remarks to the Author):

Summary:

Reviewer Comment 3.0: This article explores to what extent greenhouse gas offsetting projects are actually achieving emission reduction. To do so, the authors synthesize data from over 2,000 offsetting projects, categorize the projects into different offsetting associated sectors (e.g., forestry & land use), and estimate the average true volume in emission reduction per sector. Stunningly, the authors find that offset projects curtail much fewer emissions than the projects claim.

Overall, this is an ambitious research project that provides initial estimates of the effectiveness of carbon offset projects. Moreover, it documents differences between the offsetting sectors, and highlights aspects often neglected in offsetting projects, which is clearly of interest for the audience of Nature Communications. Personally, I think that the results could stimulate an important discussion about the credibility of carbon credits.

Despite this, I have several concerns regarding this article in its current form. I think most of the issues can be resolved.

Author Response 3.0: We thank the reviewer for the extremely helpful and constructive feedback.

Introduction:

Reviewer Comment 3.1: It is not clear what is meant with “field interventions” when the term is introduced in the introduction section (Line 87). I would provide the audience with a brief description of what is considered a field intervention (the description in the SI helped me a lot in that regard).

Author Response 3.1: We thank the reviewer for this comment. We have now made explicit in the introduction what we mean by field intervention (p.4): “We complement studies that directly evaluated carbon crediting projects with studies that evaluated similar interventions without issuing carbon credits (which we call ‘field interventions’).

Reviewer Comment 3.2: What is the problem when the real offsetting reduction is dramatically overestimated? Please provide some implications for society so that the audience knows why this paper is relevant to the general public, which is currently missing in the introduction section.

Author Response 3.2: We thank the reviewer for this comment. Based on your comments, we have reframed the introduction to detail the wider societal implications. In the first paragraph of the introduction, we now detail the wide scope of carbon crediting schemes across domestic and international mechanisms. We then discuss what constitutes high-integrity crediting mechanisms and how existing mechanisms have fallen short, including the societal implications (p. 3 and 4):

“Carbon pricing has become a central approach to mitigating climate change, though the operationalisation and geographic scope vary considerably¹. Carbon pricing has taken three approaches: emissions trading schemes (ETS), carbon taxes, and carbon crediting mechanisms. Carbon crediting mechanisms – the focus of this study – allow project developers to earn carbon credits through voluntary mitigation projects such as forest protection or renewable energy projects. These carbon crediting mechanisms are established and operated by international organisations, such as the Clean Development Mechanism (CDM) and Joint Implementation (JI) established under the Kyoto Protocol^{2,3}, national or sub-national governments, such as California’s Compliance Offset Program^{4,5}, and non-governmental entities, such as Verra and the Gold Standard Foundation^{1,6–9}. Carbon credits are used in different ways: in compliance markets⁴, countries and firms buy credits to meet targets under the Kyoto Protocol and Paris Agreement or to meet obligations under ETSs, carbon taxes or the Carbon Offsetting and Reduction Scheme for International Aviation (CORSA). In voluntary markets, governments, firms, non-governmental organisations, or individuals buy carbon credits to meet voluntary goals, such as offsetting residual emissions. Other forms of results-based finance also create demand for carbon credits, in which governments and international organisations purchase carbon credits from mitigation projects that countries implement to achieve their goals under the Paris Agreement¹.

To assess the climate benefits of carbon mitigation projects, it needs to be verified whether projects are additional and whether emission reductions or removals have been conservatively quantified, permanent, and not double counted. Additionality refers to the principle that a mitigation activity would not have occurred without the revenue from the sale of carbon credits^{7,10–12}. Conservative quantification refers to approaches that reasonably ensure that emission reductions or removals are not overestimated^{10,11}. Non-permanence refers to the risk that the

emission reductions or removals be reversed later on, for example through wildfires in forestry projects^{10,11,13}. Lastly, double counting means that an emission reduction or removal should be used only once to achieve a mitigation goal or target^{10,11,14}. Next to these basic principles, several other aspects are commonly considered important for quality. This includes avoiding negative environmental and social impacts, such as impacts on biodiversity and local communities; appropriate distribution of mitigation benefits and carbon credit revenues; ensuring that carbon mitigation projects effectively contribute to achieving net zero emissions by mid-century and avoiding locking-in carbon-intensive technologies or practices; and adequate governance structures of carbon crediting programs, including concerning transparency and third-party auditing^{3,6,9-11,15}.

Yet, carbon credits have come under considerable criticism due to growing evidence suggesting that many projects may significantly overestimate their emissions benefits or might not lead to actual emission reductions at all^{2-9,15-22} and that some projects lead to environmental or social harm⁹. Carbon credits are issued based on standards developed by carbon crediting mechanisms. The quality of carbon credits hinges on the robustness of these standards and choices made by project developers. Potential issues compromising additionality and quantification include flexibility for project developers to pick favourable data or make unrealistic assumptions^{2,3,6,8,9,24}, adverse selection^{4,13}, and use of outdated data or inappropriate methodological approaches in the standards^{2,3,6,9,11,15,16,22,25}. There is also considerable debate on the appropriateness of claims made in association with carbon credits and whether the use of carbon credits hinders or accelerates mitigation efforts.”

Reviewer Comment 3.3: Methods: By now, I do not sufficiently understand on how the true average carbon emissions reductions are estimated in Figure 3. I would add equations to the method section that sheds light on this.

Author Response 3.3: We thank the reviewer for this comment. We have substantially revised the methods sections, including all five key formulas used to compute the offset achievement ratio (p.28-32):

“Effect size extraction, transformation, and integration into quantitative framework

The central goal of our review is to assess the achieved emission reductions of carbon crediting projects relative to the issued credits, which we call the offset achievement ratio. Not all carbon

credits are used for offsetting (results-based climate finance and contribution claims made by companies are other uses), but since most credits are used for offsetting, we use the term OAR:

$$\text{OAR} = \frac{A \times (I)}{C} \quad (1)$$

where C is the number of carbon credits issued to the project, A is the additionality factor with $A \in \{0, 1\}$ (i.e., a project is either additional or not) and I are the greenhouse gas emission reductions or removals expressed in tonnes of CO₂ equivalent achieved by the project. I can be further disaggregated into:

$$I = B - (P + L) \quad (2)$$

where B is the baseline emissions, P is the project emissions, L is the leakage emissions in tonnes of CO₂e. Note that we omit other relevant factors for project impact here, such as permanence and double counting, which can further affect I . The methodological approaches to estimate baseline, project and leakage emissions depend on the project type. For instance, for cookstove projects, key factors are the number of days a stove is in use, usage rates, the fraction of non-renewable biomass, and the efficiency of the old and new stove.

To extract and standardise the estimates from individual studies, we first differentiate between carbon crediting project studies and those that do not issue credits. For field interventions that did not issue carbon credits, we extract all relevant categories according to our codebook, but only qualitatively discuss the results in the discussion section. For carbon crediting projects, we extracted and commonly did further analysis to integrate the results into our quantitative OAR framework (see Supplementary Table 6).

For the carbon crediting projects, the metric we are interested in is achieved emission reductions I that a project achieved relative to the issued credits. However, most studies do not report I but a metric correlated with emission reductions, such as deforestation rates for avoided deforestation projects, harvesting and disturbance rates for IFM projects, and biomass use for cookstoves projects. In these cases, we transform the results into emission reductions (apart from studies for which no difference can be observed between carbon crediting projects and control groups, e.g., if a cookstoves project did not lead to reductions in biomass consumption, such as Aung et al¹⁷.) In our sample, the transformation only becomes relevant for projects seeking to avoid deforestation and chemicals (as the studies on other project types covered by our review

either directly report achieved CO₂e-emission reductions or find no additionality or emissions savings; see Supplementary Table 6 for a description).

For projects seeking to avoid deforestation, we compute how changes in deforestation rates between the project area and control areas translate to hectares of land prevented from deforestation and then multiply this quantity with the carbon stored per hectare (as reported directly by projects):

$$C_{\text{total}} = R_{\text{deforest}} \times C_{\text{ha}} \quad (3)$$

where C_{total} = Total credits issued to the project, R_{deforest} = Total number of hectares prevented from deforestation, C_{ha} = Carbon stored per hectare. where separate studies on different quality elements were hard to combine, we focus our analysis on the most important factor as a lower bound of over-crediting and then describe the results of the other studies in the discussion section. For instance, as outlined in the discussion section, Bomfim et al.²⁶ evaluate project developers' estimates of the carbon per hectare in protected forests. The authors found that project estimates were 23% to 30% higher than values drawn from scientific literature. Yet, we do not consider this potential additional source of overestimation in our OAR calculation, as more research would be needed to ascertain the carbon rates per hectare on a project level.

For chemical processes, we use data on the historical waste gas generation from periods prior to carbon crediting and periods in which the plants were not eligible for crediting or in which they could not claim more credits from increasing waste gas generation to compute the OAR. In the case of SF₆ waste gas abatement, the study also compares the observed waste gas generation with default values from the Intergovernmental Panel on Climate Change (IPCC). We use the average value from these scenarios as the central estimate, determined as follows:

$$\text{OAR} = \left(\frac{\text{Likely waste gas production without crediting}}{\text{Waste Gas Production During Crediting}} \right) \times \text{Credits Issued During Study Period} \quad (4)$$

We then synthesise the individual, project-level estimates into our central estimates presented in Figure 4 via the following formula for each project type (e.g., avoided deforestation, cookstoves):

$$OAR_{pt, \text{weighted}} = \frac{\sum_{i=1}^n (OAR_{it} \times C_{it})}{\sum_{i=1}^n C_{it}} \quad (5)$$

Where OAR_i is the Offset Achievement Ratio for project i , C_i is the number of carbon credits issued to project i over the study period t , n is the total number of projects within the carbon project types pt . “

Reviewer Comment 3.4: Next, I have a substantial concern regarding whether the estimates are unbiased (i.e., of Figure 3). In meta-analyses, one typically controls for differences in the studies. For instance, one could use a fixed effects or a random effects model. I think a random effects model would be more meaningful in this analysis, since it can substantially account for heterogeneity in the studies (e.g., sample number, subjects) that affect the calculated true offsetting reduction. This, in turn, is also helps, since one could derive confidence intervals for the different sectors, acknowledging the uncertainty in heterogeneity (see, for an example, Möser & Bamberg, 2008 or Whitburn et al., 2019). Similarly, I think that confidence intervals—that one can extract from random effects models—are more meaningful than reporting the upper limits that the studies mention. Why? I would prefer an estimation of an upper limit average effect, considering the uncertainty in the studies. In doing so, you could also compare the point estimates of the different sectors by testing for statistical difference.

Author Response 3.4: We appreciate your insightful comments and the opportunity to clarify our methodological choices, particularly regarding the calculation of estimates presented in Figure 3. Your concern about the potential for bias in these estimates and the suggestion to use a random effects model to control for study heterogeneity is well-taken.

However, our systematic review faced the complexity of substantial heterogeneity in study designs, metrics, and reported outcomes. To standardize the estimates from the studies in line with our offset achievement ratio, we i) applied rigorous inclusion criteria (detailed in Supplementary Table 1, pasted below) and then ii) post-processed these estimates to make them comparable across studies.

Supplementary Table 1: Inclusion and exclusion criteria for studies.

	Population	Intervention	Comparator	Outcome	Study type
Inclusion	-	Voluntary, project-based activities that seek to reduce or remove emissions	Projects, land, or households that were not subject to the intervention (this can include historical data of the same project before it became a carbon mitigation project)	CO ₂ e-emissions reduction (or comparable metric, such as deforestation)	Quantitative estimates based on randomised controlled trial or rigorous observational studies (which includes both modelling and empirical studies). These include working paper aimed at peer-reviewed journals and PhD theses
Exclusion	-	Non-voluntary activities (e.g., mandatory regulation) or non-project-based activities (e.g., carbon tax)	Without comparator	Without quantified impact of intervention	Qualitative studies

To post-process the results in step ii), we follow the approach detailed on p. 30/31:

“For the carbon crediting projects, the metric we are interested in is achieved emission reductions / that a project achieved relative to the issued credits. However, most studies do not report / but a metric correlated with emission reductions, such as deforestation rates for avoided deforestation projects, harvesting and disturbance rates for IFM projects, and biomass use for cookstoves projects. In these cases, we transform the results into emission reductions (apart from

studies for which no difference can be observed between carbon crediting projects and control groups, e.g., if a cookstoves project did not lead to reductions in biomass consumption, such as Aung et al¹⁷.) In our sample, the transformation only becomes relevant for projects seeking to avoid deforestation and chemicals (as the studies on other project types covered by our review either directly report achieved CO₂e-emission reductions or find no additionality or emissions savings; see Supplementary Table 6 for a description).”

The nature of our data precluded a meta-analytic approach, including both fixed effects and random effects models. However, meta-regressions produce a weighted mean of the effects across studies. Our approach is similar in that we calculate a weighted mean, which is based on the individual point estimates from studies weighted by the issued credits. Hence, studies that evaluate more credits, have more weight in our overall estimates. To fully make transparent the estimates from each study and the weight, we added Supplementary Table 7 to detail each effect size (i.e., OAR) and weight. We also added the individual estimates to Figure 4 and the weighted variance of each study. Hence, while we do not calculate a meta-regression, our estimates follow the same underlying logic.

Reviewer Comment 3.5: • To evaluate the true effect of field interventions, the authors pick other projects that are suitable for comparison to the respective field intervention. To do so, the authors randomly shuffle the list of offsetting projects (Line 621). Later, when evaluating an intervention, suitable projects are taken from this list. My concern here is that it’s not clear how the randomization has influenced the results and whether they are stable when shuffling the list again. I would expect some sensitivity analysis demonstrating that the results are consistent among different randomization runs.

Author Response 3.5: We thank the reviewer for this comment. We have decided to remove the synthetic offset achievement ratio as the estimates from field interventions cannot be easily made comparable with carbon crediting projects. We have, therefore, opted to discuss the results from field interventions qualitatively, but not to use them to calculate OARs.

Reviewer Comment 3.6: I absolutely share the view of the authors that it is necessary to control for “natural” changes in the absence of offsetting projects. So, changes associated with offsetting projects (e.g., higher efficiency due to new cook stoves) may happen independently of offsetting

projects over time. Still, I would be nice to have an estimate of the emission savings ignoring such natural changes (like most offsetting projects do). I suppose that estimate would make it really tangible to the audience why a “clean” calculation (considering ex-post counterfactual reduction), like you have presented, is needed. At the same time, this might also help to make the results more credible. For instance, by now, it is not clear to me why renewable energy projects are said to have 0% actual carbon reduction (Figure 3). To what extent is this influenced by taking the “ex-post counterfactual reduction” perspective that you introduce in this manuscript?

Author Response 3.6: We thank the reviewer for this comment. Based on your comment, we have introduced two major changes. First, we detail on p. 10-12 why a counterfactual approach presented here is needed and elaborate a conceptual framework what the studies consider when calculating the achieved emissions and what is omitted by these studies:

“Carbon project developers quantify emissions reductions in line with standards and methodologies developed by carbon crediting mechanisms such as the Verified Carbon Standard by Verra. Following an audit by an accepted third party, carbon credits are issued into a registry⁸. Yet, these standards and methodologies vary in their robustness and often allow for activities to be credited that would have happened regardless of the offset programme^{2,24}, and provide flexibility to project developers to select methodological approaches and data that maximise credit issuance^{6,9}. It is, therefore, critical to contrast the emissions reduction estimates used to determine credit issuance to those achieved based on rigorous academic assessments.

We introduce the term “offset achievement ratio”, which compares studies’ quantitative estimates of carbon crediting projects’ emissions reductions with those made by project developers to generate carbon credits. An offset achievement ratio of 50% indicates that the academic literature estimates that only half of the emissions reductions claimed by project developers – and issued as carbon credits – were likely achieved. We complement these quantitative estimates with qualitative discussion of other studies including other qualitative and quantitative studies of the quality of offset methodologies and studies that assess field interventions that did not issue carbon credits but may still hold important insights on additionality, conservative quantification, or other relevant factors.

To quantify the offset achievement ratio, we rely on academic studies that evaluate voluntary, project-based activities that seek to reduce emissions or enhance removals (see Supplementary Table 1 for inclusion and exclusion criteria). We excluded studies that evaluate

non-voluntary activities such as mandatory regulations or non-project-based activities (e.g., other forms of carbon pricing such as carbon taxes). We focus on studies that evaluate project impact against a credible comparator. This comparator can include projects, land, or households that were not part of the carbon crediting projects^{4,5,7,8,17,20}; this can include historical data of the same project before it became a carbon crediting project^{15,16}. The comparator can also be values from the scientific literature⁶. For example, some studies compare individual factors used by carbon crediting projects, such as the share of users that adopt a fuel-efficient cookstove, against the body of knowledge in the published literature⁶. Studies also need to include a quantitative assessment of greenhouse gas emissions changes or a comparable environmental metric, such as deforestation rates^{7,8}. Lastly, we only include studies that use randomised controlled trials or rigorous observational studies (which construct a plausible control group⁸ or science-based comparator⁶ to estimate project impacts). The included studies fall into several categories: peer-reviewed articles¹⁷, papers aimed at peer-reviewed journals (e.g., working papers)¹⁸, and chapters in PhD theses¹⁹, which also undergo an academic examination process. We exclude qualitative studies from our quantitative assessment.

In determining the offset achievement ratio, this paper considers additionality and conservative quantification. The latter encompasses project, baseline, and leakage emissions. Figure 3 illustrates which of these issues have been addressed by the 14 studies on carbon crediting projects that were considered in determining the offset achievement ratio. Not all studies address all factors that affect a particular source of over-crediting. For instance, Aung et al.¹⁷ study the impact of fuel-efficient cookstoves on firewood usage in households that received the stove and those that did not (i.e., project and baseline emissions). However, the authors do not address other over-crediting factors related to the project emissions and baseline, such as the fraction of non-renewable biomass used to compute credit issuance. In contrast, Gill-Wiehl et al.⁶ cover all relevant factors relating to over-crediting from baseline and project emissions.

Sources of under- and over-crediting analysed in studies covered by review				
	Additionality \oplus	Conservative quantification \oplus		Permanence & No Double Issuance
		Project emissions	Baseline	Leakage
Wind power	Chan 2015	No	No	No
Forestry				
Avoided deforestation	No	West 2020, 2023 Guizar-Coutiño 2023	West 2020, 2023 Guizar-Coutiño 2023	No
IFM	Stapp 2023 Coffield 2022	Stapp 2023 Coffield 2022	Stapp 2023 Coffield 2022	Yes but no quantitative assessment
Cookstoves	No	Gill-Wiehl 2024 Aung 2016	Gill-Wiehl 2024 Aung 2016	No
Chemical				
SF6 destruction	No	Schneider 2015	Schneider 2015	No
HFC-23 destruction	No	Schneider 2011, 2015	Schneider 2011, 2015	No
Not analysed or could not be integrated into our quantitative assessment framework				
Legend	Source of over/under-crediting not covered in study	Name of study covering specific source (e.g., additionality) of over/under-crediting		Source of over/under-crediting fully covered in the study

Figure 1: Sources of under/over-crediting analysed by carbon-crediting studies covered in our review. The name of the authors shows the study which analysed the specific source of over/under-crediting, otherwise, the box indicates “No”. The figure excludes several studies that analyse offset quality, namely Calel et al.¹⁸, Badgley et al.²⁷, Holm et al.²³ and Bomfim et al.²⁶ because they could not be integrated into our quantitative assessment framework but the findings are reviewed in the discussion section. Only the first author’s name is shown due to space constraints. Note that the figure does not cover the field interventions as these did not issue carbon credits and therefore could not be integrated into our quantitative framework.”

Second, for each sub-section, we now present a detailed discussion of each study that underlies the individual results. We also clarified both in Figure 4 and the manuscript as a whole (e.g., abstract), that for some sectors (wind, improved forest management) no statistically significant effects were found in the underlying studies (p.18/19):

“Two studies^{18,19} investigated 1,966 wind power projects registered under the CDM in India and China. These studies only investigate the additionality of these projects. Globally, around half of credits from wind power projects were issued under the CDM, 63% of which were generated in China. We use only the data by Chan and Huenteler¹⁹ to estimate the offset achievement ratio of wind power projects, because Calel et al.¹⁸ only provide an upper bound for additionality and the authors make clear that the results could be as low as zero.

Chan and Huenteler¹⁹ investigated the additionality of 2,051 wind projects, of which 1,494 were financed in China under the CDM between 2007-2012. They found no statistically significant evidence that projects that received funding from the CDM were less financially viable than those that were constructed without support. However, they show that projects under the CDM used more foreign technologies and larger wind turbines, potentially increasing technology transfer. In addition, they document a small positive effect on CDM projects being sited in previously undeveloped areas. Yet, these positive effects can only be ascribed to CDM financing if projects were additional, which appears not to be the case.

Calel et al.¹⁸ investigate the additionality of 1,350 wind projects in India, of which 472 were financed under the CDM between 2000 and 2013. They developed a new conceptual framework called Blatantly Infra-marginal Projects (BLIMPs), which identifies particularly obvious cases of non-additionality. The approach allows the authors to identify projects that were less financially attractive but were built even without selling carbon credits. For around half of these projects, they identified that these projects had lower capacity factors, were in less windy locations and were sited further away from electrical substations, and hence overall likely to be less financially attractive than the CDM projects.

The authors indicate that low additionality is likely due to the capital intensity of this project type. Utility-scale renewable energy projects require high up-front investments and a secure cash flow to secure funding from banks and investors²⁸. As revenue streams from selling carbon credits are often low in comparison to revenues from electricity sales and carbon credit prices may fluctuate substantially, as in the CDM, revenues generated by carbon credits are unlikely to affect the financial viability of renewable energy projects substantially^{19,24}. ”

Minor:

Reviewer Comment 3.8: • I do not understand “reduce deforestation from deforestation” in the introduction section (Line 100). Please revise.

Author Response 3.8: We thank the author for this comment. We have removed the line.

Reviewer Comment 3.9: Line 542: To me, it is vague what “similar, real-world projects” could be here. Provide some specifications when projects are similar.

Author Response 3.9: We thank the author for this comment. As we have now shifted the focus away from synthetic offset achievement ratios based on field interventions, we have removed the line.

References:

- Whitburn, J., Linklater, W. and Abrahamse, W. (2020), Meta-analysis of human connection to nature and proenvironmental behavior. *Conservation Biology*, 34: 180-193. <https://doi.org/10.1111/cobi.13381>
- Möser, G., & Bamberg, S. (2008). The effectiveness of soft transport policy measures: A critical assessment and meta-analysis of empirical evidence. *Journal of Environmental Psychology*, 28:1, 10-26.

Reviewer #4 (Remarks to the Author):

Reviewer Comment 4.0: This manuscript addresses an important issue in the field of carbon offset projects. In the recent past, particular attention has been paid to offset projects in the forestry sector. This manuscript makes a significant contribution by providing a scientifically sound analysis of the actual emission reductions of projects in different sectors. A quantitative approach is used to show that the emission reductions claimed by projects are not actually achieved. The article complements existing studies with a novel scientific approach. The results show the need to fundamentally rethink carbon offset projects.

The scientific method used is sound and fully meets the expected requirements. The description of the scientific methods allows the traceability. In my opinion, there are no fundamental flaws. In some places the manuscript should be revised.

Author Response 4.0. We thank the reviewer for the extremely helpful and constructive feedback.

Reviewer Comment 4.1: Line 80: the review tool AS Review should be described briefly, possibly in the methods section.

Author Response 4.1: We now state in the methods section (p.29): “We relied on the AI-supported systematic review tool AS Review⁹ to order our complete study set in order of probable relevance. AS Review is a software tool that allows for more efficient screening of titles and abstracts. By labelling a set of potentially relevant articles AS Review prioritises articles to be investigated for relevance in the screening process. Screening articles without prioritisation is error-prone and inefficient as only a small fraction of articles is relevant.”

Reviewer Comment 4.2: Line 142 ff: obviously the reference to the figures is not correct. Here reference is made to figure 2 and not to figure 1.

Author Response 4.2: We thank the reviewer for this comment. We have removed the line.

Reviewer Comment 4.3: line 170 ff. It should be described in more detail which biophysical and socio-economic characteristics are subject to the comparison. Forests differ in terms of tree

species composition, site factors (e.g., climate, soil, topography), previous management activities, biomass volume, and other critical factors. The limitation to the shown aspects (forest type, distance to roads and distance to forest edge) are not sufficient for a selection of reference areas.

Author Response 4.3: We thank the reviewer for this comment. Based on your comment, we decided to provide a more encompassing explanation of the comparator. As we cover 6 different project types, we decided to broaden the scope of our explanation (p. 10/11) rather than focusing on a single example:

“To quantify the offset achievement ratio, we rely on academic studies that evaluate voluntary, project-based activities that seek to reduce emissions or enhance removals (see Supplementary Table 1 for inclusion and exclusion criteria). We excluded studies that evaluate non-voluntary activities such as mandatory regulations or non-project-based activities (e.g., other forms of carbon pricing such as carbon taxes). We focus on studies that evaluate project impact against a credible comparator. This comparator can include projects, land, or households that were not part of the carbon crediting projects^{4,5,7,8,17,20}; this can include historical data of the same project before it became a carbon crediting project^{15,16}. The comparator can also be values from the scientific literature⁶. For example, some studies compare individual factors used by carbon crediting projects, such as the share of users that adopt a fuel-efficient cookstove, against the body of knowledge in the published literature⁶. Studies also need to include a quantitative assessment of greenhouse gas emissions changes or a comparable environmental metric, such as deforestation rates^{7,8}. Lastly, we only include studies that use randomised controlled trials or rigorous observational studies (which construct a plausible control group⁸ or science-based comparator⁶ to estimate project impacts). The included studies fall into several categories: peer-reviewed articles¹⁷, papers aimed at peer-reviewed journals (e.g., working papers)¹⁸, and chapters in PhD theses¹⁹, which also undergo an academic examination process. We exclude qualitative studies from our quantitative assessment.”

Reviewer Comment 4.4: line 312-347: a major problem in surveying carbon stocks in forests is the accuracy of the survey method. This should be addressed because the uncertainties that measurement, reporting and verification systems are subject to can have a decisive influence on the quantification of carbon stocks (see e.g. <https://doi.org/10.1016/j.ecolecon.2019.106513>)

Author Response 4.4: We thank the reviewer for this comment. We now detail on p.24 the risks associated with quantification:

“Yet, West et al. and Guizar-Coutinho do not assess the assumptions project developers make regarding the carbon contained in the forest areas, which can further lead to over-crediting (see Figure 3). Bomfim et al.²⁶ assess project developers’ estimates of the carbon per hectare in protected forests. If these estimates are overstated, then the issuance of credits will also be inflated. Based on a representative sample of 12 projects across four key VCS methodologies, the authors show that project developers have significant leeway in assessing carbon content in forests. They found that project estimates were 23% to 30% higher than values drawn from scientific literature. We do not consider this potential additional source of overestimation in our OAR calculation, as more research would be needed to ascertain the carbon rates per hectare on a project level.”

Reviewer Comment 4.5: Line 576: CBL instead of BL?

Author Response 4.5: We thank the reviewer for this comment. We have fully updated the relevant sections.

Reviewer Comment 4.6: Line 586 ff: It is unclear whether the average values shown for the effectiveness index will be used for further study. If so, the choice of average values could mask extremes in individual projects.

Author Response 4.6: We thank the reviewer for this comment. We have now made explicit the underlying estimations used to calculate the offset achievement ratio (OAR) across sectors. For instance, Figure 4a now contains both the average OAR across project types, and the individual project-level estimates (Figure 4b). The Figure also contains the exact geographic scope, crediting mechanism, and underlying studies (identified by colour codes). In addition, Supplementary Table 7 contains a detailed list with the study name, estimate, and weight (i.e., issued carbon credits). Lastly, we have included a detailed dataset that allows readers to identify the exact data points used to calculate the OAR.

Reviewer Comment 4.7: Line 606-607: E is not defined. Should it be EI?

Author Response 4.7: We thank the reviewer for this comment. We have fully updated the relevant sections.

Reviewer Comment 4.8: Line 624: It would be interesting to get an example from forestry projects, instead of the rather straightforward example of cooking stoves.

Author Response 4.8: We thank the reviewer for this comment. We have decided to remove the synthetic offset achievement ratio as the estimates from field interventions cannot be easily made comparable with carbon crediting projects. We have, therefore, opted to discuss the results from field interventions qualitatively, but not to use them to calculate OARs.

Reviewer #5 (Remarks to the Author):

Reviewer Comment 5.0: In general, this is a well-written manuscript. The authors attempted to reveal the implementation efficiency of carbon offset projects across the main sectors that have been thoroughly discussed by previous studies. For a bibliometric study, it always has a cutting-edge perspective, together with methodological challenges. With no exception, I have following concerns towards this manuscript too:

Author Response 5.0: We thank the reviewer for the extremely helpful and constructive feedback.

Reviewer Comment 5.1: First and most importantly, I understand it is very difficult to obtain actual, real data regarding ex-ante and ex-post carbon offset levels/volume. However, according to the authors' claim, "the offset achievement ratio" is the share of achieved emissions reductions based on credible academic studies relative to the claims made by project developers' ex-ante". I did not see many details here about how the authors defined "credible academic studies" and how these studies would provide plausible results that can be used to match and compare with projects. Therefore, such a ratio appears to be a subjective judgment based on a subjective material and some results are skeptical such as "0% for renewable energy" which is counterfactual. Unless the authors provide a strong defense, the true value of this manuscript is less likely to be explored.

Author Response 5.1: We thank the reviewer for this comment. We now clearly explain in the manuscript and in Supplementary Table 1 what the inclusion and exclusion criteria are (p. 10/11):

"To quantify the offset achievement ratio, we rely on academic studies that evaluate voluntary, project-based activities that seek to reduce emissions or enhance removals (see Supplementary Table 1 for inclusion and exclusion criteria). We excluded studies that evaluate non-voluntary activities such as mandatory regulations or non-project-based activities (e.g., other forms of carbon pricing such as carbon taxes). We focus on studies that evaluate project impact against a credible comparator. This comparator can include projects, land, or households that were not part of the carbon crediting projects^{4,5,7,8,17,20}; this can include historical data of the same project before it became a carbon crediting project^{15,16}. The comparator can also be values from the scientific literature⁶. For example, some studies compare individual factors used by carbon crediting

projects, such as the share of users that adopt a fuel-efficient cookstove, against the body of knowledge in the published literature⁶. Studies also need to include a quantitative assessment of greenhouse gas emissions changes or a comparable environmental metric, such as deforestation rates^{7,8}. Lastly, we only include studies that use randomised controlled trials or rigorous observational studies (which construct a plausible control group⁸ or science-based comparator⁶ to estimate project impacts). The included studies fall into several categories: peer-reviewed articles¹⁷, papers aimed at peer-reviewed journals (e.g., working papers)¹⁸, and chapters in PhD theses¹⁹, which also undergo an academic examination process. We exclude qualitative studies from our quantitative assessment.“

These criteria are also detailed in Supplementary Table 1:

Supplementary Table 1: Inclusion and exclusion criteria for studies.

	Population	Intervention	Comparator	Outcome	Study type
Inclusion	-	Voluntary, project-based activities that seek to reduce or remove emissions	Projects, land, or households that were not subject to the intervention (this can include historical data of the same project before it became a carbon mitigation project)	CO2e-emissions reduction (or comparable metric, such as deforestation)	Quantitative estimates based on randomised controlled trial or rigorous observational studies (which includes both modelling and empirical studies). These include working paper aimed at peer-reviewed journals and PhD theses
Exclusion	-	Non-voluntary activities (e.g., mandatory regulation) or non-project-based activities (e.g., carbon tax)	Without comparator	Without quantified impact of intervention	Qualitative studies

Second, for each sub-section, we now present a detailed discussion of each study that underlies the individual results. We also clarified both in Figure 4 and the manuscript as a whole (e.g., abstract), that for some sectors (wind, improved forest management) no statistically significant effects were found in the underlying studies (p.18/19):

“Two studies^{18,19} investigated 1,966 wind power projects registered under the CDM in India and China. These studies only investigate the additionality of these projects. Globally, around half of credits from wind power projects were issued under the CDM, 63% of which were generated in China. We use only the data by Chan and Huenteler¹⁹ to estimate the offset achievement ratio of wind power projects, because Calel et al.¹⁸ only provide an upper bound for additionality and the authors make clear that the results could be as low as zero.

Chan and Huenteler¹⁹ investigated the additionality of 2,051 wind projects, of which 1,494 were financed in China under the CDM between 2007-2012. They found no statistically significant evidence that projects that received funding from the CDM were less financially viable than those that were constructed without support. However, they show that projects under the CDM used more foreign technologies and larger wind turbines, potentially increasing technology transfer. In addition, they document a small positive effect on CDM projects being sited in previously undeveloped areas. Yet, these positive effects can only be ascribed to CDM financing if projects were additional, which appears not to be the case.

Calel et al.¹⁸ investigate the additionality of 1,350 wind projects in India, of which 472 were financed under the CDM between 2000 and 2013. They developed a new conceptual framework called Blatantly Infra-marginal Projects (BLIMPs), which identifies particularly obvious cases of non-additionality. The approach allows the authors to identify projects that were less financially attractive but were built even without selling carbon credits. For around half of these projects, they identified that these projects had lower capacity factors, were in less windy locations and were sited further away from electrical substations, and hence overall likely to be less financially attractive than the CDM projects.

The authors indicate that low additionality is likely due to the capital intensity of this project type. Utility-scale renewable energy projects require high up-front investments and a secure cash flow to secure funding from banks and investors²⁸. As revenue streams from selling carbon credits are often low in comparison to revenues from electricity sales and carbon credit prices may fluctuate substantially, as in the CDM, revenues generated by carbon credits are unlikely to affect the financial viability of renewable energy projects substantially^{19,24}.”

Reviewer Comment 5.2: Second, the discussion of leakage, durability, and co-benefits is important, but such a discussion appears to be inconsistent with the main focus of this study i.e. the carbon offset mismatch.

Author Response 5.2: We thank the reviewer for this comment. We agree and based on your comment introduced the following changes. First, in the introduction we now mention these quality characteristics, including leakage, durability, and co-benefits, but make clear that we focus on additionality and conservative quantification (p.3):

“To assess the climate benefits of carbon mitigation projects, it needs to be verified whether projects are additional and whether emission reductions or removals have been conservatively quantified, permanent, and not double counted. Additionality refers to the principle that a mitigation activity would not have occurred without the revenue from the sale of carbon credits^{7,10-12}. Conservative quantification refers to approaches that reasonably ensure that emission reductions or removals are not overestimated^{10,11}. Non-permanence refers to the risk that the emission reductions or removals be reversed later on, for example through wildfires in forestry projects^{10,11,13}. Lastly, double counting means that an emission reduction or removal should be used only once to achieve a mitigation goal or target^{10,11,14}. Next to these basic principles, several other aspects are commonly considered important for quality. This includes avoiding negative environmental and social impacts, such as impacts on biodiversity and local communities; appropriate distribution of mitigation benefits and carbon credit revenues; ensuring that carbon mitigation projects effectively contribute to achieving net zero emissions by mid-century and avoiding locking-in carbon-intensive technologies or practices; and adequate governance structures of carbon crediting programs, including concerning transparency and third-party auditing^{3,6,9-11,15}.”

But then we state of p.4: “In this paper, we focus our analysis on the two most basic principles of carbon credit integrity: additionality and conservative quantification.”

Based on your comment, we have also added a figure that clearly explains the focus of our review and the underlying study (Figure 3, pasted below):

Sources of under- and over-crediting analysed in studies covered by review						
	Additionality \oplus	Conservative quantification \oplus			Permanence & No Double Issuance	
		Project emissions	Baseline	Leakage		
Wind power	Chan 2015	No	No	No	Not analysed or could not be integrated into our quantitative assessment framework	
Forestry						
Avoided deforestation	No	West 2020, 2023 Guizar-Coutiño 2023	West 2020, 2023 Guizar-Coutiño 2023	No		
IFM	Stapp 2023 Coffield 2022	Stapp 2023 Coffield 2022	Stapp 2023 Coffield 2022	Yes but no quantitative assessment		
Cookstoves	No	Gill-Wiehl 2024 Aung 2016	Gill-Wiehl 2024 Aung 2016	No		
Chemical						
SF6 destruction	No	Schneider 2015	Schneider 2015	No		
HFC-23 destruction	No	Schneider 2011, 2015	Schneider 2011, 2015	No		
Legend	Source of over/under-crediting not covered in study	Name of study covering specific source (e.g., additionality) of over/under-crediting		Source of over/under-crediting only partially covered by study		Source of over/under-crediting fully covered in the study

Figure 2: Sources of under/over-crediting analysed by carbon-crediting studies covered in our review. The name of the authors shows the study which analysed the specific source of over/under-crediting, otherwise, the box indicates “No”. The figure excludes several studies that analyse offset quality, namely Calel et al.¹⁸, Badgley et al.²⁷, Holm et al.²³ and Bomfim et al.²⁶ because they could not be integrated into our quantitative assessment framework but the findings are reviewed in the discussion section. Only the first author’s name is shown due to space constraints. Note that the figure does not cover the field interventions as these did not issue carbon credits and therefore could not be integrated into our quantitative framework.

Reviewer Comment 5.3: Third, I would not say the discussion is not logic or the evidence provided by the authors is not rich. However, this is again, a commonly seen issue for a review style study. The manuscript is entitled “systematic review”, but in fact the authors attempt to further find out some reasons why there is a gap between proposals and reality. This is a pure economic logic (i.e., data, model and analysis) which is different to be presented in this type of studies thus it is not surprising that to some extent, the discussion is not quite comprehensive, even sometimes the authors say, “we could only find two empirically rigorous studies...”.

Author Response 5.3: We thank the reviewer for this comment. Based on your comment, we have fully restructured and reworked the discussion section to be firmly grounded in the evidence we review. Please see the main manuscript for the discussion section, but here as an example the cookstoves discussion section (p.20-21):

“Cookstoves

Information from two studies^{6,17} investigating 52 projects was used to estimate an average offset achievement ratio of 10.7% (Supplementary Table 6 explains how we post-process and synthesise the results from these studies; this is the weighted average across projects covered by studies). Aung et al.¹¹ assess project and baseline emissions for one CDM project. Gill-Wiehl et al.⁶ analysed 51 projects (40% of all issued credits across independent crediting mechanisms from 5 key methodologies) and assessed all relevant factors (apart from additionality and leakage) in the quantification of emission reductions, including fraction of non-renewable biomass, adoption/usage rates, and emissions factors. Distributing fuel-efficient cookstoves seeks to reduce greenhouse gas emissions by encouraging households in low- and middle-income countries to switch to a less GHG-intensive fuel or a more energy-efficient stove. Most cookstove projects are registered under the Gold Standard (GS), the VCS or the CDM and rely on methodologies from GS and the CDM⁶.

Aung et al.¹⁷ ran a randomised controlled trial to evaluate the climate impacts of one CDM-approved stove replacement project in India. The author team randomly assigned 187 households to either receive a fuel-efficient replacement (96 households) for their traditional stove or to serve as a control group. Overall, Aung et al. find no statistically significant impact on fuelwood usage between the intervention and control groups (hence, we assume an OAR of 0%). They document that 40% of households that received the fuel-efficient stove continued using the traditional stove.

They hypothesise that the lack of reductions might also be due to households cooking larger meals with the improved stoves (“rebound effect”), thereby eliminating any efficiency-based reductions in fuelwood consumption.

While Aung et al. only analysed one project, Gill-Wiehl et al.¹⁷ assessed the overall quality of a substantial portion of cookstove credits on the voluntary carbon market, covering 51 projects, five key cookstove methodologies and a comprehensive set of factors. The authors recalculate the likely emission reductions of these analysed cookstove projects by scrutinising key methodological assumptions made to issue credits. Overall, the authors find that the project sample likely only achieved 10.9% of the claimed emission reductions, though there is a large variation between methodologies (please note that the OAR of 10.7 calculated for the overall project type is the weighted average by issued credits from Gill-Wiehl et al.⁶ and Aung et al.¹⁷). For instance, Gold Standard’s Metered methodology²⁹, which assesses fuel use directly, features the lowest over-crediting risks of all methodologies.

Hence, while efficient cookstoves have been found to offer considerable sustainable development benefits, the literature suggests that their low carbon credit quality is due to a lack of rigour and flexibility in how methodologies allow projects to (1) determine the fraction of non-renewable sources of fuelwood and other biomass (fNRB), (2) assess actual use of the new and old stoves, and (3) translate these values into changes in fuel consumption. For adoption, usage, and stacking rates, all methodologies except Gold Standard’s Metered methodology allow projects to use infrequent and simple surveys that are vulnerable to social desirability and recall bias (challenges in remembering past usage). Further, there are limitations in how methodologies allow projects to estimate fuel consumption. The project stove’s efficiency is often determined in laboratory settings that are highly artificial and inapplicable to real-world conditions. While these approaches are better than surveys, they are still vulnerable to overestimation from the Hawthorne effect (when stove users change their behaviour because they are observed)³⁰.

In addition, numerous other studies have evaluated one or a few factors in the emission reduction calculation and compared them to carbon crediting projects or methodologies’ approaches, finding over-crediting from the choice of fNRB³¹ and methods to track adoption/usage rates³² and under-crediting from emission factors³³. Rigorous evaluations of field interventions have found substantial variation in the achieved emission reductions^{34–38}, which are rarely on par with the levels claimed by carbon crediting projects⁶. Studies investigating the additionality and leakage of cookstove projects are still nascent in the literature but analysing these factors would be important to fully assess the achieved emission reductions⁶.”

Reviewer Comment 5.4: Finally, the conclusion is too simple. The authors should not only state that they identified a gap, but they should also provide solid references and materials towards research directions in future. What should the academic world do for bridging such a gap?

Author Response 5.4: We thank the reviewer for this comment. Based on your comment, we have fully redrafted the conclusion section. We now clearly identify important future research directions based on our review:

“Conclusion

We synthesized the extant literature relying on experimental or rigorous observational methods, covering 14 studies on 2,420 carbon mitigation projects and 51 studies investigating similar field interventions implemented without issuing carbon credits. Our analysis covers about one-fifth of the credit volume issued to date, almost 1 billion tons. We estimate that less than 16% of the carbon credits issued to the investigated projects constitute real emission reductions, with 11% for cookstoves, 16% for SF₆ destruction, 25% for avoided deforestation, 68% for HFC-23 abatement, and no statistically significant emission reductions from wind power projects in China and improved forest management projects in the United States.

Our review, therefore, documents substantial and systemic quality problems across all analysed project types, which further strengthens the evidence by previous cross-cutting analyses of the CDM and the JI^{2,39}. Carbon credits are issued based on standards developed by carbon crediting mechanisms. The quality of carbon credits hinges on the robustness of these standards, the choices made by project developers in applying these standards, and the thoroughness of the checks by third-party auditors and the carbon crediting mechanism. Our review highlights that many project developers pick favourable data or make unrealistic assumptions⁶. Some methodologies make use of outdated data or inappropriate methodological approaches⁴, which can lead to adverse selection³⁵ or perverse incentives^{12,15,16}. Our results also indicate that there is substantial heterogeneity across project types and methodologies.

The reviewed studies suggest that existing approaches to assess additionality have led to many non-additional projects being registered. To address this issue, carbon crediting programs could limit eligibility to project types that have a high likelihood of additionality and of being

effectively supported by revenues from carbon credits. For example, following criticism regarding additionality, Verra and the Gold Standard excluded wind power projects in most countries from eligibility. However, newer crediting mechanisms, such as the Global Carbon Council, include these projects in their scope. This change would result in a much narrower set of eligible project types.

Our findings also suggest that the standards and methodologies to quantify emission reductions need to be considerably improved. Such improvements should address a range of issues, in particular reducing project developers' flexibility in making favourable methodological assumptions to maximise credit generation^{6,8,8,20}; using conservative assumptions and data based on the latest scientific evidence^{6,18,19,26}; and addressing the risk of adverse selection^{4,5} and perverse incentives^{15,16}. Carbon crediting programs may also exclude project types from eligibility where it is very difficult to ascertain whether calculated emission reductions result from the mitigation activities or exogenous factors that impact emissions, an issue that has also been referred to as 'signal-to-noise' issue.

Various other studies, not included in our analysis, suggest that quality issues also persist for many other project types not covered by our analysis^{2,3,11,39,40}. This suggests that our estimate that 812 million carbon credits do not represent actual emission reductions should be considered as a lower bound as many more credits currently traded may not constitute real emissions reductions.

In addition, questions around additionality and leakage remain only partly addressed by the literature^{24,25} and our review does not cover two other potential sources of over-crediting: permanence and double counting. For instance, Holm et al.²³ assess the non-permanence risk for 57 VCS forestry projects. Project developers need to make non-permanence risk assessments which inform the number of carbon credits set aside to insure against future reversals. Holm et al.²³ recalculate the assessments made by project developers based on the latest scientific literature and find that – on average – project developers were issued on average 26.5% more credits than an appropriate risk management would demand. Cookstoves projects also face a non-permanence risk as more fuel-efficient cookstoves lead to the preservation of carbon stocks in surrounding forests, but this risk is not accounted for by any of the carbon crediting mechanisms. Double issuance presents another risk as more than half of cookstove projects are co-located in areas where projects seek to avoid deforestation⁴¹. Hence, our estimates likely present the upper bound of OAR, which would likely be even lower if these factors were considered.

Our findings also suggest that more research is needed to better understand the quality of credits across different project types. For instance, for renewable energy, the extant literature providing quantitative assessments of achieved emission reductions focuses primarily on grid-connected wind power projects^{18,19}, though the literature on small-scale renewable energy is scant. More work is also needed to explore the full sources of over/under-crediting of projects with existing evaluations.

Demand for carbon credits is expected to grow significantly over the next decades, with increased demand from voluntary carbon market buyers, domestic compliance markets, CORSIA and countries using Article 6 of the Paris Agreement¹. Yet, our results substantiate doubts about the environmental quality of carbon credits from the project types we study. These quality issues need to be addressed for carbon crediting mechanisms to meaningfully contribute to climate change mitigation.”

References

1. World Bank. *State and Trends of Carbon Pricing 2023*. (2023). doi:10.1596/39796.
2. Cames, M. *et al.* *How Additional Is the Clean Development Mechanism?* 173
https://ec.europa.eu/clima/sites/clima/files/ets/docs/clean_dev_mechanism_en.pdf (2016)
doi:CLIMA.B.3/SERI2013/0026r.
3. Kollmuss, A., Schneider, L. & Zhezherin, V. *Has Joint Implementation Reduced GHG Emissions? Lessons Learned for the Design of Carbon Market Mechanisms*.
<https://www.sei.org/publications/has-joint-implementation-reduced-ghg-emissions-lessons-learned-for-the-design-of-carbon-market-mechanisms/> (2015).
4. Stapp, J. *et al.* Little evidence of management change in California's forest offset program. *Communications Earth and Environment* **4**, 1–10 (2023).
5. Coffield, S. R. *et al.* Using remote sensing to quantify the additional climate benefits of California forest carbon offset projects. *Global Change Biology* **28**, 6789–6806 (2022).
6. Gill-Wiehl, A., Kammen, D. M. & Haya, B. K. Pervasive over-crediting from cookstove offset methodologies. *Nat Sustain* (2024) doi:10.1038/s41893-023-01259-6.
7. West, T. A. P., Börner, J., Sills, E. O. & Kontoleon, A. Overstated carbon emission reductions from voluntary REDD+ projects in the Brazilian Amazon. *Proceedings of the National Academy of Sciences of the United States of America* **117**, 24188–24194 (2020).
8. West, T. A. P. *et al.* Action needed to make carbon offsets from tropical forest conservation work for climate change mitigation. *Science* **877**, 873–877 (2023).
9. Haya, B. K. *et al.* *Quality Assessment of REDD+ Carbon Credit Projects. Berkeley Carbon Trading Project*. <https://gspp.berkeley.edu/research-and-impact/centers/cepp/projects/berkeley-carbontrading-project/REDD+>.
10. ICVCM. Core Carbon Principles. *ICVCM* <https://icvcm.org/the-core-carbon-principles/>.
11. CCQI. The Carbon Credit Quality Initiative. <https://carboncreditquality.org> (2024).

12. Haya, B. *et al.* Managing uncertainty in carbon offsets: insights from California's standardized approach. *Climate Policy* **20**, 1112–1126 (2020).
13. Badgley, G. *et al.* California's forest carbon offsets buffer pool is severely undercapitalized. *Frontiers in Forests and Global Change* **5**, (2022).
14. Schneider, L. *et al.* Double counting and the Paris Agreement rulebook. *Science* **366**, 180–183 (2019).
15. Schneider, L. R. Perverse incentives under the CDM: An evaluation of HFC-23 destruction projects. *Climate Policy* **11**, 851–864 (2011).
16. Schneider, L. & Kollmuss, A. Perverse effects of carbon markets on HFC-23 and SF6 abatement projects in Russia. *Nature Climate Change* **5**, 1061–1063 (2015).
17. Aung, T. W. *et al.* Health and Climate-Relevant Pollutant Concentrations from a Carbon-Finance Approved Cookstove Intervention in Rural India. *Environmental Science and Technology* **50**, 7228–7238 (2016).
18. Calel, R., Colmer, J., Dechezleprêtre, A. & Glachant, M. Do Carbon Offsets Offset Carbon? *SSRN Electronic Journal* (2021) doi:10.2139/ssrn.3950103.
19. Chan, G. & Huenteler, J. Financing Wind Energy Deployment in China through the Clean Development Mechanism. in *Essays on Energy Technology Innovation Policy* (2015).
20. Guizar-Coutiño, A., Jones, J. P. G., Balmford, A., Carmenta, R. & Coomes, D. A. A global evaluation of the effectiveness of voluntary REDD+ projects at reducing deforestation and degradation in the moist tropics. *Conservation Biology* **36**, 1–13 (2022).
21. Schneider, L., Lazarus, M. & Kollmuss, A. *Industrial N2O Projects Under the CDM: Adipic Acid – A Case of Carbon Leakage?* <https://www.sei.org/publications/industrial-n2o-projects-cdm-adipic-acid-case-carbon-leakage/> (2010).
22. Haya, B. K. *et al.* Comprehensive review of carbon quantification by improved forest management offset protocols. *Frontiers in Forests and Global Change* **6**, (2023).

23. Holm, J. A., Anderegg, W. R. L., Bomfim, B., So, I. S. & Haya, B. K. Durability. in *Quality assessment of REDD+ carbon credit projects. Berkeley Carbon Trading Project*. (Berkeley Carbon Trading Project, 2023).
24. Haya, B. Carbon offsetting: An efficient way to reduce emissions or to avoid reducing emissions? An investigation and analysis of offsetting design and practice in India and China [Doctoral dissertation, Energy & Resources Group, University of California]. (2010).
25. Haya, B. K. The California Air Resources Board's U.S. Forest offset protocol underestimates leakage.
26. Bomfim, B., West, T. A. P., Holm, J. A., Anderegg, W. R. L. & Haya, B. K. Forest Carbon Accounting. in *Quality Assessment of REDD+ Carbon Credit Projects* (Berkeley Carbon Trading Project, 2023).
27. Badgley, G. *et al.* Systematic over-crediting in California's forest carbon offsets program. *Global Change Biology* **28**, 1433–1445 (2022).
28. Probst, B., Westermann, L., Anadón, L. D. & Kontoleon, A. Leveraging private investment to expand renewable power generation: Evidence on financial additionality and productivity gains from Uganda. *World Development* **140**, 105347 (2021).
29. Gold Standard Foundation. Methodology for metered & measured energy cooking devices. *Gold Standard for the Global Goals* <https://globalgoals.goldstandard.org/news-methodology-for-metered-measured-energy-cooking-devices/>.
30. Simons, A. M., Beltramo, T., Blalock, G. & Levine, D. I. Using unobtrusive sensors to measure and minimize Hawthorne effects: Evidence from cookstoves. *Journal of Environmental Economics and Management* **86**, 68–80 (2017).
31. Bailis, R., Wang, Y., Drigo, R., Ghilardi, A. & Masera, O. Getting the numbers right: revisiting woodfuel sustainability in the developing world. *Environ. Res. Lett.* **12**, 115002 (2017).

32. Ramanathan, T. *et al.* Wireless sensors linked to climate financing for globally affordable clean cooking. *Nature Clim Change* **7**, 44–47 (2017).
33. Sanford, L. & Burney, J. Cookstoves illustrate the need for a comprehensive carbon market. *Environ. Res. Lett.* **10**, 084026 (2015).
34. Bensch, G. & Peters, J. The intensive margin of technology adoption - Experimental evidence on improved cooking stoves in rural Senegal. *Journal of Health Economics* **42**, 44–63 (2015).
35. Bensch, G. & Peters, J. Alleviating deforestation pressures? Impacts of improved stove dissemination on charcoal consumption in urban senegal. *Land Economics* **89**, 676–698 (2013).
36. Brooks, N. *et al.* How much do alternative cookstoves reduce biomass fuel use? Evidence from North India. *Resource and Energy Economics* **43**, 153–171 (2016).
37. Beltramo, T. & Levine, D. I. The effect of solar ovens on fuel use, emissions and health: Results from a randomised controlled trial. *Journal of Development Effectiveness* **5**, 178–207 (2013).
38. Berkouwer, S. B. & Dean, J. T. *Credit, Attention, and Externalities in the Adoption of Energy Efficient Technologies by Low-Income Households. American Economic Review* vol. 112 (2022).
39. Spalding-Fecher, R. *et al.* *Assessing the Impact of the Clean Development Mechanism.* http://www.cdmpolicydialogue.org/research/1030_impact.pdf (2012).
40. Schneider, L. Assessing the additionality of CDM projects: practical experiences and lessons learned. *Climate Policy* **9**, 242–254 (2009).
41. Calyx Global. Cooking up Quality: Carbon credits from efficient cookstove projects face integrity issues worth fixing. <https://calyxglobal.com/blog-post?q=18> (2023).

REVIEWERS' COMMENTS

Reviewer #5 (Remarks to the Author):

Dear authors,

I am happy with those changes you have made in response to my comments. All my concerns have been successfully addressed.